# NOT ALL BITS ARE EQUAL: SCALE-DEPENDENT MEMORY OPTIMIZATION STRATEGIES FOR REASONING MODELS

**Junhyuck Kim**[k]**, Ethan Ewer**[w*]**, Taehong Moon**[k]**, Jongho Park**[b]**, Dimitris Papailiopoulos**[w,m]
[k]KRAFTON, [w]University of Wisconsin–Madison, [b]UC Berkeley, [m]Microsoft Research

## ABSTRACT

While 4-bit quantization has emerged as a memory-optimal choice for non-reasoning models and zero-shot tasks across scales, we show that this universal prescription fails for reasoning models, where KV cache rather than model size can dominate memory. Through systematic experiments on mathematical, code generation, and knowledge-intensive reasoning tasks, we find a *scale-dependent trade-off*: models with an effective size below 8-bit 4B parameters achieve better accuracy by allocating memory to larger weights, rather than longer generation, while larger models benefit from the opposite strategy. This scale threshold also determines when parallel scaling becomes memory-efficient and whether KV cache eviction outperforms KV quantization. Our findings show that memory optimization for LLMs cannot be scale-agnostic, while providing principled guidelines: for small reasoning models, prioritize model capacity over test-time compute, while for large ones, maximize test-time compute. Our results suggest that optimizing reasoning models for deployment requires fundamentally different strategies than those established for non-reasoning ones.

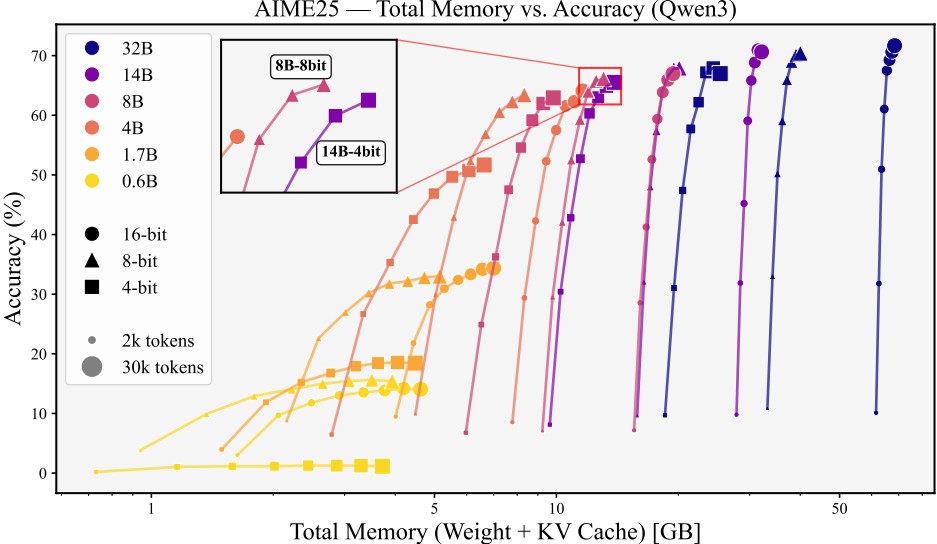

Figure 1: **Memory vs. Accuracy for serial test-time scaling on AIME25.** The plot illustrates the trade-off between pass@1 accuracy and total memory (weight + KV cache) for the Qwen3 family. Model weights are quantized to 4- and 8-bit using GPTQ. Along each curve, the KV cache grows as the generation length increases via budget forcing. For models effectively smaller than an 8-bit 4B, increasing the token budget to saturation is memory-inefficient. Furthermore, for mathematical reasoning, higher weight precision (8- and 16-bit) proves more memory-efficient than 4-bit.

---

*This work was done during an internship at KRAFTON.

# 1 INTRODUCTION

Prior memory–performance trade-off studies on Large Language Models (LLMs) for non-reasoning models have focused mostly on compressing model weights, since the model weights generally consume far more GPU memory than the Key-Value (KV) cache (Dettmers & Zettlemoyer, 2023; Frantar et al., 2022; Lin et al., 2024). Modern reasoning models, however, generate substantially more tokens, causing the proportionally increasing KV cache to become a significant bottleneck. For instance, a Qwen3-4B model with 4-bit weights occupies 2.49 GB, but its KV cache for a 32k-token generation requires 4.42 GB ($\approx 1.8\times$ the weights). This bottleneck is magnified in batched inference: with model weights amortized, the aggregated KV cache becomes the primary memory constraint. With the KV cache becoming a dominant component of memory, it is unclear whether the results established for non-reasoning models still hold for long-generation reasoning tasks.

In this work, we aim to investigate the general principles of memory compression for reasoning models. In addition to the conventional factors of *model size* and *weight precision*, our analysis incorporates three other factors that distinctly affect memory–accuracy trade-offs for reasoning models: *generation length*, *parallel scaling*, and *KV cache compression*. Overall, we ask the question:

> *Under a fixed memory budget, how should one navigate the trade-offs between* **model size**, **weight precision**, **token budget**, **sample size for parallel scaling**, *and* **KV cache compression** *to maximize performance for reasoning models?*

We conduct an empirical study mainly focusing on the Qwen3 model family (0.6B to 32B) (Yang et al., 2025) across four benchmarks: AIME25, GPQA-Diamond, LiveCodeBench, and MATH500. Furthermore, we evaluate the DeepSeek-R1-Distill (Guo et al., 2025) and OpenReasoning-Nemotron (Majumdar et al., 2025) reasoning model families to verify that our main findings from Qwen3 generalize beyond a single model family. Our investigation spans over **1,700** different scenarios, exploring 4-bit and 8-bit GPTQ weight quantization (Frantar et al., 2022), reasoning token budgets from 2k to 30k, parallel scaling via majority voting with up to 16 samples, and two approaches to KV cache compression: eviction, using R-KV (Cai et al., 2025) and StreamingLLM (Xiao et al., 2023b), and quantization with HQQ (Badri & Shaji, 2023). While our findings do not provide specific prescriptions for all tasks or models, we provide general principles to consider for memory-efficient reasoning models with minimal loss of accuracy.

**Our contributions.** In Section 4, we investigate how to allocate memory between model weights and the KV cache under serial test-time scaling. For example, which setting leads to higher accuracy: a 32B 8-bit LLM with less KV cache (*i.e.*, less test-time compute), or a 32B 4-bit LLM with more KV cache? We find that there is *no* optimal strategy that is universal across scale: for models with effective size (parameters × bits per weight) below 8-bit 4B ($\approx 4.2$ GB), allocating more memory to model weights yields larger gains, whereas above this threshold, memory is better spent increasing the test-time budget until performance saturates.

We also discover that the choice of weight precision depends on the nature of the task. For knowledge-intensive reasoning, 4-bit weight quantization is broadly memory-optimal, consistent with established findings on the effectiveness of 4-bit or lower precision for zero-shot, non-reasoning models (Dettmers & Zettlemoyer, 2023; Frantar et al., 2022; Chee et al., 2023). For mathematical reasoning and code generation tasks, however, the higher fidelity of 8 or 16-bit model weights with smaller KV cache often provides stronger performance, suggesting that intricate computational tasks are more sensitive to loss in precision.

Orthogonal to longer generations, increasing the number of generations can yield substantial gains (Brown et al., 2024), yet its memory efficiency remains under-explored. Parallel scaling via majority voting on top of serial scaling introduces another trade-off: a larger KV cache proportional to group size for higher accuracy, assuming a batched inference setting. This strategy is only more memory-efficient than serial scaling for models with an effective size at or above 8-bit 4B. Interestingly, for such models, the memory-optimal group size also increases with the total memory budget. Moreover, we show that using an external verifier such as Process Reward Model (PRM) is memory-inefficient compared to self-contained majority voting.

In Section 5, we investigate how KV cache compression affects the memory–accuracy trade-off by considering both KV cache eviction and quantization methods. Across model sizes and weight pre-

cisions, both eviction and quantization advance their Pareto frontiers beyond the baseline without cache compression. The choice of compression method should be dictated by effective size: eviction offers a better memory trade-off for small models (effective size below 8-bit 4B), while both strategies are competitive for larger models.

Overall, the memory-optimal strategy for reasoning models is *not* universal, but is instead mainly governed by the model's effective size. We summarize our main empirical findings as follows:

1. For models effectively smaller than 8-bit 4B, it is more memory-efficient to allocate memory to larger weights than to longer generations, while larger models benefit more from longer generations.

2. 4-bit weights are broadly memory-optimal for knowledge-intensive tasks, while 8-bit or 16-bit weights are more memory-efficient for mathematical reasoning and code generation.

3. Parallel scaling only improves the memory–accuracy trade-off for models effectively larger than 8-bit 4B. The memory-optimal group size increases with the memory budget.

4. Weight quantization alone is not sufficient for memory-optimal reasoning; compressing the KV cache leads to more memory-efficient reasoning.

5. KV cache eviction provides a better memory–accuracy trade-off than KV cache quantization for models with an effective size smaller than an 8-bit 4B model.

## 2 BACKGROUND

**Weight-only quantization.** Weight-only post-training quantization replaces full-precision weights with low-bit representations without retraining, reducing memory usage. Weight-only quantization allows lower bit-widths compared with weight-activation quantization, as it is more robust to quantization error (Yao et al., 2023). However, weight-only quantization requires dequantization before multiplying with activations, so it does not reduce computational cost during inference. Any speedup instead comes from reduced memory movement. In this work, we adopt GPTQ (Frantar et al., 2022), a weight-only quantization method that minimizes layer-wise quantization error using a small calibration set and updates weights using inverse-Hessian information. We additionally replicate key experiments using AWQ (Lin et al., 2024) and FP8 (Micikevicius et al., 2022) to verify that our conclusions do not depend on a specific quantization scheme.

**KV cache quantization.** KV cache quantization stores key and value tensors at reduced precision to lower the memory footprint during decoding. Unlike weight-only quantization, KV quantization is applied online: during prefill, the KV tensors are quantized and cached in low precision; during decode, they are dequantized on the fly for attention computation. Prior work conventionally maintains a small full-precision buffer for the most recent tokens, appending new entries during decoding. In this work, we use per-channel symmetric quantization of both keys and values with an HQQ backend (Badri & Shaji, 2023), a fast, calibration-free method well-suited for online KV cache quantization.

**KV cache eviction.** KV cache eviction reduces cache size and the cost of attention computation. For reasoning models, we consider dynamic eviction policies that continuously evict the KV cache during decoding. Early work, such as StreamingLLM (Xiao et al., 2023b), employs a sliding-window mechanism that preserves the most recent key and value tensors, in addition to the initial sequence tokens known as the attention sink. More recently, R-KV (Cai et al., 2025) proposes redundancy-aware selection for reasoning models: it estimates token importance and redundancy during decoding and jointly selects non-redundant, informative tokens to retain, reporting near-baseline accuracy with a small fraction of the KV cache. In Section 5, we study how these eviction policies, together with KV cache quantization, shift the trade-off frontiers.

**Test-time scaling.** We scope this work to test-time scaling methods that do not rely on external models such as verifiers or process reward models. Reasoning models are typically trained to produce an extended chain-of-thought, continuing generation with planning and reflection to improve performance (Guo et al., 2025; Jaech et al., 2024; Yang et al., 2025); we refer to this as serial scaling. Muennighoff et al. (2025) introduces budget forcing to scale serial responses beyond the model's

natural length: when the model attempts to stop, a short cue is appended to continue decoding to a specified token budget. Another line of work, parallel scaling, generates multiple independent reasoning trajectories (Brown et al., 2024). In its simplest form, without any external model, majority voting selects the final answer as the most frequent among the independently sampled outputs (Wang et al., 2022). Further related work is discussed in Appendix A.

## 3 EXPERIMENTAL SETUP

We systematically explore the memory–accuracy trade-offs by measuring how accuracy and memory footprints are affected by five key factors: the number of parameters ($N$), weight precision ($P_W$), test-time token budget ($T$), sampling group size ($G$, with $G > 1$ indicating multiple samples for majority voting), and KV cache compression strategy ($\pi_{\mathrm{kv}}$, e.g., eviction or quantization).

The memory cost is given by

$$M \;=\; M_{\mathrm{weights}}(N, P_W) \;+\; M_{\mathrm{kv}}(N, \pi_{\mathrm{kv}}, T, G),$$

where $M_{\mathrm{weights}}$ is the memory footprint of the weights, roughly proportional to $N \cdot P_W$. Note that throughout the paper, we use model *size* to refer to the number of parameters $N$ and *effective size* or *scale* to refer to the memory footprint of the weights, $M_{\mathrm{weights}}$. $M_{\mathrm{kv}}$ is the KV cache memory, which is roughly proportional to $N$, $G$, and $T$, except when $\pi_{\mathrm{kv}} = $ eviction, where the cost becomes constant beyond a certain token budget. Please refer to Appendix B for the exact memory cost equations and Table 1 for model-specific values.

Table 1: **Memory footprints of the Qwen3 model family.**

| Model | Model Weight (GB) | | | KV Cache (GB) | | | |
|---|---|---|---|---|---|---|---|
| | 4-bit | 8-bit | 16-bit | 2k tokens | 18k tokens | 30k tokens | 30k tokens × 16 samples |
| Qwen3-0.6B | 0.50 | 0.71 | 1.40 | 0.21 | 1.92 | 3.20 | 51.27 |
| Qwen3-1.7B | 1.26 | 1.93 | 3.78 | 0.21 | 1.92 | 3.20 | 51.27 |
| Qwen3-4B | 2.49 | 4.19 | 7.49 | 0.27 | 2.47 | 4.12 | 65.91 |
| Qwen3-8B | 5.68 | 8.94 | 15.26 | 0.27 | 2.47 | 4.12 | 65.91 |
| Qwen3-14B | 9.30 | 15.50 | 27.51 | 0.31 | 2.75 | 4.58 | 73.24 |
| Qwen3-32B | 18.01 | 32.66 | 61.02 | 0.49 | 4.39 | 7.32 | 117.19 |

**Models.** We experiment with the Qwen3 model family (Yang et al., 2025) (0.6B–32B), which offers a wide range of model sizes for a fine-grained systematic study. We additionally evaluate DeepSeek-R1-Distill (Guo et al., 2025) and OpenReasoning-Nemotron (Majumdar et al., 2025) to test whether our findings generalize beyond Qwen3.

**Tasks.** Experiments are conducted on challenging benchmarks representing complementary difficulty profiles. AIME25 (AIME, 2025) is a competition-level mathematical benchmark that stresses multi-step reasoning, and MATH500 (Lightman et al., 2023) extends this to a larger, more diverse math set. In contrast, GPQA-Diamond (Rein et al., 2024) emphasizes scientific knowledge and integrated reasoning across domains such as chemistry, biology, and physics (Li et al., 2025b), while LiveCodeBench (Jain et al., 2024) evaluates reasoning in code generation.

**Inference details.** Unless otherwise specified, we report accuracy averaged over 32 generations per instance and sample with temperature 0.6. Following Muennighoff et al. (2025), for serial scaling with budget forcing, if generation terminates earlier than the desired token budget, we replace the end-of-sequence token with the prompt `Wait` and continue decoding until the target budget is reached. When the desired budget is met, we inject the prompt `**Final Answer**\n\\boxed{`. We evaluate token budgets from 2k to 30k in 4k increments. Our code is available at `https://github.com/krafton-ai/not-all-bits-are-equal`.

## 4 TEST-TIME SCALING WITH WEIGHT-ONLY QUANTIZATION

*When aiming for the best performance under limited memory, how should memory be allocated between model weights and KV cache? Additionally, when allocating space for model weights, is it better to use more parameters at lower precision or fewer parameters at higher precision?*

To answer these questions, we study test-time scaling across different model sizes ($N$) and weight precisions ($P_W \in \{4, 8, 16\}$) by varying the test-time token budget ($T$). We use GPTQ to quantize models to 4- and 8-bit precision. For this analysis, we fix $\pi_{\mathrm{kv}}$ to keep all cache entries (no eviction, full precision) and first present results for a sampling group size of $G = 1$. We later discuss parallel scaling with $G > 1$ and other $\pi_{\mathrm{kv}}$ policies.

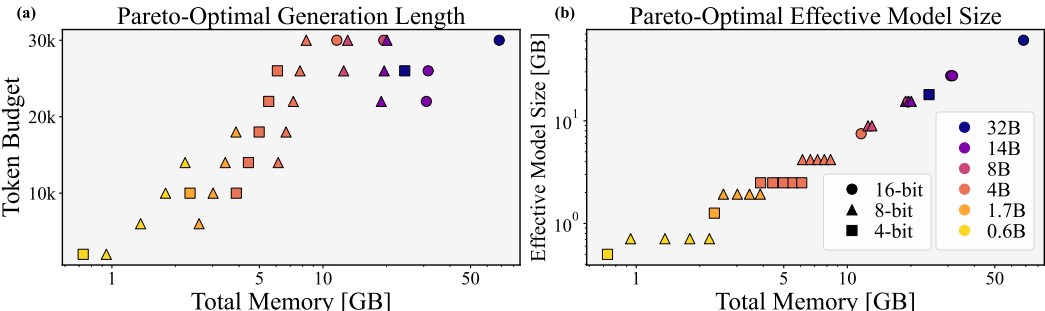

Figure 2: **Composition of Pareto-optimal configurations (AIME25, Qwen3).** The token budget (a) and effective model size (b) are plotted against the total memory budget for configurations on the Pareto frontier from Figure 1. The plots illustrate a strategic shift: at lower memory budgets ($<10\,\mathrm{GB}$), increasing effective model size is memory-efficient, whereas at higher budgets, increasing the token budget becomes the dominant strategy for improving performance.

Figure 1 reveals the Pareto frontier for accuracy versus total memory under serial scaling with a full-precision KV cache. Analyzing the configurations that lie on this frontier provides practical recommendations for optimizing model selection, weight precision, and test-time budgets within fixed memory constraints:

**For models effectively smaller than 8-bit 4B, memory is better spent on increasing the effective model size rather than increasing the test-time budget until saturation.** While extending the generation budget of a small model is often viewed as a way to trade higher latency for lower memory usage compared to using a large model, our analysis reveals that this is a false economy. In fact, for models effectively smaller than 8-bit 4B, this strategy is often suboptimal in total memory. Figure 2 shows that for memory budgets below $8\,\mathrm{GB}$, the Pareto frontier is advanced primarily by increasing model size, not the token budget. For instance, the 1.7B model in 8-bit with a 6k token budget outperforms the 0.6B model in 8-bit with an 18k token budget. Similarly, the 4B model in 4-bit with a 10k token budget surpasses the 1.7B model in 8-bit with an 18k token budget, demonstrating that choosing a model with a larger effective size is better under a similar memory budget. As our latency analysis confirms (Appendix C.1), these configurations with larger effective sizes are also faster because end-to-end latency is dominated by the token budget, making the choice to increase the model's effective size strictly dominant.

**For large models with an effective size at or above 8-bit 4B, memory is more efficiently used when increasing the test-time budget until performance saturates.** In direct contrast to the strategy for small models, extending the generation budget is a more memory-efficient way to improve accuracy for large models. This strategic shift is clearly illustrated in Figure 2, where for memory budgets larger than $10\,\mathrm{GB}$, the best-performing configurations on the Pareto frontier consistently feature token budgets above 20k. In this regime, increasing the token budget becomes the dominant method for improving accuracy.

We further confirm that our conclusions about weight precision are not tied to GPTQ. In Appendix C.2, we show that AWQ and FP8 weight quantization yield nearly identical memory–accuracy curves to GPTQ (Figure 12), and that the observed trends are robust across quantization schemes. Separately, while the above analysis assumes a scenario where each inference instance uses the entire model and KV cache, in practice, model weights can be amortized across multiple concurrent generations, fundamentally changing the memory dynamics; Appendix C.3 analyzes this setting under different theoretical batch sizes.

> **Finding 1**
>
> The memory-efficient allocation strategy between model weights and KV cache is scale-dependent. For models effectively smaller than 8-bit 4B, memory is more efficiently allocated to increasing the effective model size. For models at or above this threshold, it becomes more memory-efficient to increase the test-time budget until performance saturates.

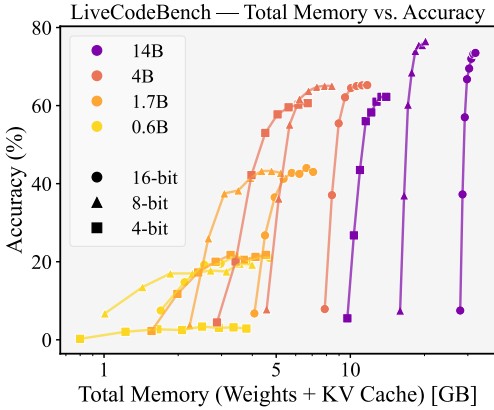

Figure 3: **Memory vs. Accuracy on Live-CodeBench (Qwen3).** Higher precision (8-/16-bit) remains more memory-efficient than 4-bit. The memory-optimal strategy shifts from favoring model weights at small budgets to longer generations at larger budgets.

Figure 4: **Memory vs. Accuracy on GPQA-Diamond (Qwen3).** Unlike mathematical reasoning and code generation, 4-bit weights remain broadly memory-optimal for this knowledge-intensive task across memory budgets.

**The memory-optimal weight precision is task- and size-dependent.** Our findings show that for mathematical reasoning tasks, 4-bit weight quantization is consistently memory-inefficient. On the AIME25 benchmark, 8-bit is memory-optimal for small models ($N \in \{0.6B, 1.7B\}$), as the performance gains from reallocating memory saved by 4-bit quantization to a larger token budget are insufficient to compensate for the accuracy loss. This inefficiency of 4-bit persists at larger $N$, where 8-bit and 16-bit configurations achieve higher accuracy at comparable memory. This is shown in Figure 2 (b), where 8-bit or 16-bit weights are most often memory-optimal along the frontier for memory budgets larger than 6 GB. Notably, the 8B model in 8-bit consistently outperforms the 14B model in 4-bit (Figure 1), and the 32B model in 4-bit is strictly dominated by both the 14B model in 8-bit and the 8B model in 16-bit. Such findings are in direct contrast to Dettmers & Zettlemoyer (2023). LiveCodeBench exhibits a similar preference for higher weight precision over 4-bit (Figure 3), and we refer to Appendix C.4 for a detailed analysis of LiveCodeBench and MATH500. However, we do find that for knowledge-intensive tasks, 4-bit quantization is broadly memory-optimal. As shown in Figure 4 for GPQA-Diamond, the frontier shifts to favor lower precision. This suggests that different task types place different demands on model parameters. Mathematical reasoning may rely on numerical precision within the weights, which is damaged by aggressive 4-bit quantization. On the other hand, knowledge-intensive tasks prioritize maximizing the number of parameters to increase knowledge capacity, making large 4-bit models more memory-efficient.

> **Finding 2**
>
> For knowledge-intensive tasks, 4-bit is broadly memory-optimal. For mathematical reasoning and code generation tasks, higher precision is required. 8-bit is memory-optimal for small models ($N \in \{0.6B, 1.7B\}$), while both 8-bit and 16-bit are competitive at larger numbers of parameters.

In addition to serial scaling by increasing the token budget, we can introduce a parallel scaling axis by increasing the sampling group size ($G$). Assuming a batched inference setting, the KV cache grows with $G$, in exchange for higher accuracy. This raises another key question:

*When is it more memory-efficient to allocate memory to parallel samples, versus allocating it to a larger effective model size or a longer token budget?*

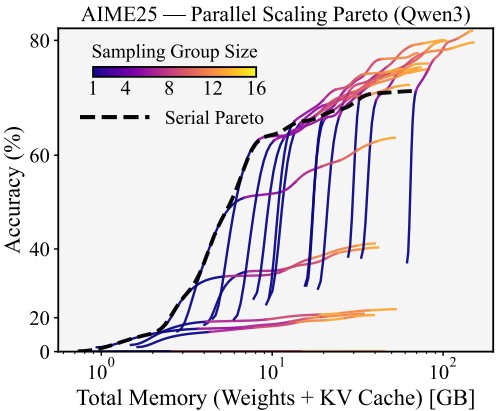

Figure 5: **Effect of parallel scaling on the Pareto frontier (Qwen3).** Each colored curve represents the Pareto frontier for a specific model size and weight precision, obtained by increasing the sampling group size, $G$. Pareto frontier for serial scaling ($G = 1$) across all models is shown as the dotted line. Parallel scaling is only effective for large models.

Figure 6: **Effect of parallel scaling on the Pareto frontier (DeepSeek-R1-Distill).** A similar scale-dependent pattern holds for DeepSeek-R1-Distill: parallel scaling is memory-inefficient for small models but improves the Pareto frontier for sufficiently large ones.

**The effectiveness of parallel scaling is scale-dependent.** For systematic evaluation, we use budget forcing to control the token budget for each of the $G$ parallel samples and use majority voting to select the final answer. This majority voting protocol corresponds to self-consistency with majority aggregation (Wang et al., 2022). Figure 5 shows how parallel scaling affects the memory–accuracy trade-off. The dotted line marks the Pareto frontier from serial scaling alone. Each colored curve represents the frontier for a specific model configuration as the group size, $G$, is increased (see Appendix C.5, Figure 15 for a per-model breakdown). For models effectively smaller than 8-bit 4B, parallel scaling is memory-inefficient, as its configurations lie below the frontier established by serial scaling alone. However, for large models, parallel scaling improves the trade-off, and the memory-optimal group size $G$ on the global Pareto frontier increases with the memory budget. While group sizes of $4 \leq G < 8$ are memory-optimal in the 16.4–28.9 GB range, for budgets above 28.9 GB, the frontier is pushed by even larger groups ($G \geq 8$). This scale-dependent behavior is not unique to Qwen3: for DeepSeek-R1-Distill and OpenReasoning-Nemotron (Figures 6 and 16), parallel scaling is memory-inefficient for smaller effective sizes but improves the Pareto frontier once models are sufficiently large. Appendix C.6 provides detailed results on these reasoning model families.

> **Finding 3**
>
> For models effectively smaller than 8-bit 4B, serial scaling alone provides a better memory–accuracy trade-off than parallel scaling. For models effectively larger than this, parallel scaling improves the trade-off, and the memory-optimal group size $G$ on the global Pareto frontier increases with the memory budget.

While this work focuses on memory trade-offs, practical scenarios also consider latency and throughput constraints. We analyze these trade-offs in Appendix C.1.

## 4.1 Parallel Scaling with an External Verifier

We evaluate Best-of-N parallel scaling with ActPRM-X (Duan et al., 2025) as an external PRM, accounting for its 7B (13.28 GB) memory overhead in our total memory budget. Comparing the Pareto frontiers formed by serial scaling, majority-vote parallel scaling, and PRM-based Best-of-N (Figure 7), we find that the external verifier is consistently memory-inefficient: accuracy gains are marginal relative to the substantial fixed memory cost, and in low-memory regimes it can even underperform serial scaling. These results suggest that under tight memory budgets, self-contained strategies such as majority voting are preferable to relying on large external verifiers.

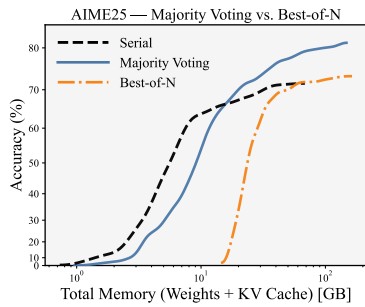

Figure 7: **Parallel scaling with Best-of-N using ActPRM-X.**

## 5 Test-Time Scaling with Weight and KV Cache Compression

Our analysis so far shows that while allocating more tokens generally improves accuracy, it is not always memory-efficient, especially for effectively small models where the KV cache can dominate total memory. While compressing the KV cache via quantization or eviction can reduce this footprint, it comes at a potential accuracy cost. This raises the following question:

*How do KV cache compression strategies—eviction and quantization—alter the overall memory–accuracy trade-off, and which approach leads to stronger reasoning?*

To answer this, we evaluate both compression strategies across model sizes and weight precisions. For eviction, we use R-KV with target KV budgets of 2k, 4k, and 8k tokens. For KV cache quantization, we use symmetric per-channel quantization to 2-, 4-, and 8-bit precisions with a group size of 64 and a full-precision residual buffer of 128 tokens. The results are averaged over 8 generations per instance. We first show that both methods are broadly beneficial and then provide a detailed analysis to determine which strategy is optimal under different conditions.

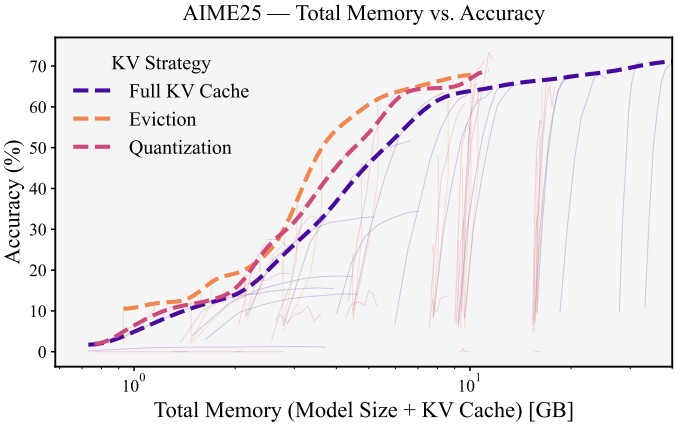

Figure 8: **Memory vs. Accuracy by KV cache compression strategy (AIME25, Qwen3).** The plot shows the Pareto frontiers of KV cache compression across model sizes and weight precisions under serial scaling with budget forcing. Eviction uses R-KV with token budgets of 2k, 4k, and 8k. Quantization is symmetric per-channel (group size 64) at 2-, 4-, and 8-bits. Faint background lines show curves for individual (model size, weight precision, KV strategy) configurations. Both compression strategies consistently improve the memory–accuracy trade-off.

**KV cache eviction and quantization consistently advance the Pareto frontier across all tested model sizes and weight precisions.** Our first key finding, illustrated in Figure 8, is that the aggregate Pareto frontiers for both quantization and eviction decisively advance beyond a baseline without

compression for models with 4-bit, 8-bit, and 16-bit weights. This improvement demonstrates that these strategies enable either higher accuracy at the same memory budget or the same accuracy at a lower memory cost, regardless of the model weight precision. The benefits are especially pronounced in the low-memory regime below 10 GB, where smaller models are most constrained by the KV cache. This indicates that even when model weights are aggressively compressed, the KV cache contains significant redundancies that can be exploited. Our results, therefore, establish KV cache compression as an essential and broadly beneficial strategy for the memory-efficient deployment of reasoning models.

> **Finding 4**
>
> Weight quantization alone is not sufficient for memory-optimal reasoning. KV cache compression advances the memory–accuracy frontier across all weight precisions.

Having established that KV cache compression is broadly beneficial, we now analyze which compression strategy, quantization or eviction, is preferable for a given model size $N$ and weight precision $P_W$. Figure 9 shows the resulting memory–accuracy trade-offs, where each strategy shapes the curves differently. Quantization reduces the memory cost per token, shifting the curves leftward, typically with some accuracy degradation. Eviction, in contrast, enforces a fixed memory ceiling for the KV cache, resulting in characteristic vertical curves where accuracy improves while memory usage remains constant.

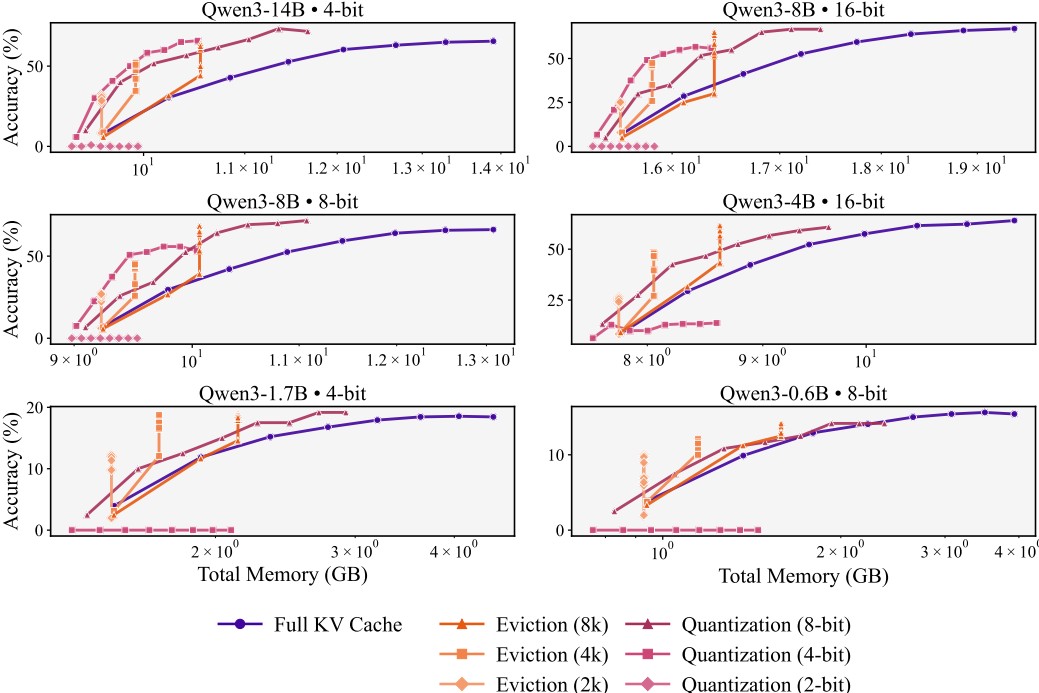

Figure 9: **Per-model Memory vs. Accuracy by KV cache strategy (AIME25).** Each plot illustrates the memory–accuracy trade-off for a single model size and weight precision, comparing a full KV cache baseline against R-KV eviction and symmetric per-channel quantization. Points along each curve represent increasing number of processed tokens via budget forcing.

**Eviction is more effective than quantization for small models.** For models with an effective size smaller than an 8-bit 8B model, eviction consistently provides the best memory–accuracy trade-off. As shown in Figure 9 for the full-precision 4B model, eviction with an 8k token budget maintains near-lossless in maximum accuracy while substantially reducing total memory. This observation holds across all weight precisions for the 4B model (see Appendix C.7, Figure 18 for these results).

In contrast, aggressive 4-bit KV cache quantization causes a significant drop in accuracy at these small effective sizes. This suggests that effectively small models are more sensitive to the numerical errors introduced by quantization, whereas eviction preserves the full precision of a smaller, more critical set of tokens. For instance, on the 1.7B model with 4-bit weight precision, eviction achieves the best memory trade-off while maintaining high accuracy, whereas an 8-bit quantized KV cache, while effective, requires significantly more memory to reach a similar performance level.

**Quantization becomes competitive with eviction for large models.** For models with an effective size larger than an 8-bit 8B model, the clear advantage of eviction diminishes as quantization becomes a highly competitive strategy. On the 8B model with 16-bit weights, for example, quantization and eviction achieve comparable memory–accuracy trade-offs. While 4-bit KV cache quantization is competitive, eviction with smaller budgets (4k or 2k) offers a similar trade-off in low-memory regimes. This suggests that large models, with their greater number of effective parameters, are more robust to the precision loss from quantization. However, we find that more aggressive 2-bit quantization still results in a significant loss of accuracy.

> **Finding 5**
>
> KV cache eviction provides a better memory–accuracy trade-off than KV cache quantization for models with an effective size smaller than an 8-bit 8B model. For models at or above this threshold, quantization becomes an increasingly competitive strategy.

## 6  CONCLUSION

Under real-world circumstances with fixed memory budgets, deploying reasoning models is ultimately a problem of *where* to spend bytes, and practitioners are presented with a myriad of choices. Our work reformulates test-time scaling around this constraint. We study the trade-offs in allocating memory across model size, weight precision, KV cache compression, token budget, and sampling group size for reasoning models. We find that the memory-optimal inference strategy for reasoning models *cannot be a one-size-fits-all prescription*: instead, it depends on the model's capacity (determined by effective size) and the nature of the task.

For smaller model sizes (typically models under the 8B size), prioritizing model weights yields better memory–accuracy trade-offs by using higher-precision 8-/16-bit weights for mathematical reasoning and favoring KV cache eviction over quantization. For larger models, increasing the token budget until saturation and leveraging parallel scaling become the dominant strategies. Importantly, the inflection point where extra KV cache beats extra model weight may change as models become more sophisticated. However, by shifting the focus from FLOPs-based test-time scaling laws to practical memory constraints, our framework and analysis provide general principles for deploying reasoning models effectively.

## 7  LIMITATIONS AND FUTURE WORK

Our scope is intentionally focused to keep the search space tractable and inference-only. For testtime scaling, we rely on prompt injection for serial scaling and majority voting for parallel scaling, and only include a limited evaluation of an external verifier rather than a comprehensive comparison of verifier-based methods. We compare a small set of post-training quantization schemes but do not systematically study alternative KV cache eviction algorithms or training-time approaches such as quantization-aware training. Our main analysis centers on the Qwen3 family, chosen for its broad size range and fixed architecture, and two challenging benchmarks (AIME25 for mathematical reasoning and GPQA-Diamond for knowledge-intensive reasoning). However, additional experiments on the DeepSeek-R1-Distill and OpenReasoning-Nemotron model families and on the LiveCodeBench and MATH500 benchmarks suggest that our findings generalize beyond a single model family and a single pair of benchmarks. These choices were necessary to maintain a tractable search space, which already spans over 1,700 experimental configurations, and focus on self-contained inference strategies, leaving a broader comparison of methods as a clear avenue for future work.

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

## A    RELATED WORK

**Train-time scaling and knowledge capacity.**    Foundational scaling studies (Kaplan et al., 2020; Henighan et al., 2020; Hoffmann et al., 2022) establish power-law relationships between model size, data, and loss, yielding prescriptions for compute-optimal training under fixed compute budgets. While these results provide guidance for allocating parameters and tokens during pre-training, they do not consider inference-time compute and hence require new extrapolations (Gadre et al., 2025). In parallel, capacity-oriented analyses estimate what models can store, either by modeling knowledge as information per parameter or by measuring memorization versus generalization (Morris et al., 2025; Allen-Zhu & Li, 2024). These views motivate a budget-centric view but leave precision and inference-time trade-offs under deployment constraints unspecified. Bit-normalized studies examine how performance at different precision scales with total model bits (Dettmers & Zettlemoyer, 2023) or the amount of training data (Kumar et al., 2024), particularly in zero-shot or few-shot scenarios. Feng et al. (2025); Mekala et al. (2025) further show that reduced numerical precision can markedly impair arithmetic reasoning and long-context performance unless compensated by a larger model size, indicating interactions between precision, task structure, and context length.

**Inference-time methods and scaling laws.**    Chain-of-thought prompting elicits intermediate steps, and self-consistency improves performance by sampling diverse rationales and aggregating them via majority voting (Wei et al., 2022; Wang et al., 2022). Modern reasoning models are trained to generate substantially more tokens, yielding significant gains across benchmarks (Wang et al., 2022; Wei et al., 2022; Brown et al., 2024; Muennighoff et al., 2025; Guo et al., 2025; Yang et al., 2025; Comanici et al., 2025; Jaech et al., 2024; Qwen Team, 2025; Kimi Team, 2025). Test-time scaling laws study how performance changes with increased FLOPs, tokens, or number of generations, comparing strategies such as majority voting, best-of-n, and verification-based search (Brown et al., 2024; Wu et al., 2024; Snell et al., 2024; Muennighoff et al., 2025; Sadhukhan et al.; Wang et al., 2025; Zhao et al., 2025). However, these studies do not capture the impact of compression techniques such as weight-only quantization, which reduces memory and latency without affecting FLOPs. While concurrent works by Liu et al. (2025b) and Kurtić et al. (2025) study quantization in reasoning models, their focus is on accuracy degradation rather than memory trade-offs. Our work is distinct in its memory-centric view: we analyze the trade-offs in allocating a fixed memory budget between model weights and test-time compute (generation length and parallelism), incorporating the full cost of the KV cache. We also broaden the scope of compression techniques to include KV cache eviction.

**Efficient inference.**    Various strategies have been proposed to address challenges in LLM quantization, particularly handling outliers (Frantar et al., 2022; Xiao et al., 2023a; Lin et al., 2024; Kim et al., 2023; Dettmers et al., 2022). Quantization-aware training extends this idea by training models with quantized weights in the forward pass (Liu et al., 2023a; Ma et al., 2024; Liu et al., 2025c). Post-training KV cache compression techniques can be categorized into eviction and quantization. Eviction methods selectively discard less important entries based on different criteria (Cai et al., 2025; Xiao et al., 2023b; Zhang et al., 2023; Liu et al., 2023b; Li et al., 2024; Ge et al., 2023), while quantization approaches reduce the precision of cached values (Badri & Shaji, 2023; Kang et al., 2024; Liu et al., 2024; Kim et al., 2024; Hooper et al., 2024).

**Quantization and reasoning.**    Recent work has also examined how compression and low-bit quantization affect reasoning. Li et al. (2025a) find that aggressive weight quantization notably harms mathematical reasoning at low precision, consistent with our observation that 4-bit precision is memory-inefficient for mathematical reasoning. Liu et al. (2025b) show that the impact varies by bit-width and model family, and Zhang et al. (2025) benchmark compressed reasoning models on complex tasks to chart accuracy under compression. Our work complements these studies by framing the problem as selecting a memory-optimal strategy for reasoning, identifying a scale-dependent threshold for allocating memory between model weights and longer generations, and incorporating KV cache compression into this analysis.

## B  MEMORY EQUATIONS AND SPECIFICATIONS

The total memory cost $M$ is the sum of the memory required for the model weights $M_{\text{weights}}$ and the KV cache $M_{\text{kv}}$.

**Weight Memory.**   The total memory footprint for weights is the sum of memory for the quantized and unquantized parameters. The general equation is

$$M_{\text{weights}} \approx \underbrace{\left( N_{\text{quant}} \cdot \frac{P_W}{8} + \frac{N_{\text{quant}}}{g_W} \cdot \frac{P_S + P_Z}{8} \right)}_{\text{Quantized Parameters}} + \underbrace{\left( N_{\text{unquant}} \cdot \frac{P_{\text{native}}}{8} \right)}_{\text{Unquantized Parameters}} \quad \text{[bytes]},$$

where $N_{\text{quant}}$ and $N_{\text{unquant}}$ are the number of quantized and unquantized parameters, respectively, $P_W$ is the low-bit precision for weights, $g_W$ is the group size, $P_S$ and $P_Z$ are the bit-widths for the scales and zero-points, and $P_{\text{native}}$ is the native precision of the unquantized layers.

In our specific setup using GPTQ, the large linear layers are quantized, while components such as the token embedding matrix, normalization layer weights, and the final language model head remain in native BF16 precision. For our experiments, we use a group size $g_W = 128$, a scale precision of $P_S = 16$ (FP16), and symmetric quantization, making the zero-point precision $P_Z = 0$.

**KV Cache Memory.**   Without compression, the KV cache memory is given by

$$M_{\text{kv}} = G \cdot T \cdot n_{\text{layers}} \cdot n_{\text{kv\_heads}} \cdot d_{\text{head}} \cdot 2 \cdot \frac{P_{\text{native}}}{8} \quad \text{[bytes]},$$

where $G$ is the sampling group size, $T$ is the number of tokens, $n_{\text{layers}}$ is the number of layers, $n_{\text{kv\_heads}}$ is the number of Key/Value heads, $d_{\text{head}}$ is the dimension per head, the factor of 2 accounts for both Key and Value tensors, and $P_{\text{native}}$ is the native precision of the cache elements in bits (e.g., 16 for BF16).

The memory cost is modified by different KV cache strategies:

- **Eviction:** This strategy reduces the number of tokens stored. The memory cost is

$$M_{\text{kv}} = G \cdot \min(T, T_{\text{retain}}) \cdot n_{\text{layers}} \cdot n_{\text{kv\_heads}} \cdot d_{\text{head}} \cdot 2 \cdot \frac{P_{\text{native}}}{8},$$

  where $T_{\text{retain}}$ is the maximum number of tokens retained by the policy. In our experiments, we use R-KV and test $T_{\text{retain}} \in \{8192, 4096, 2048\}$.

- **Quantization:** This strategy reduces the precision but introduces overhead for quantization parameters. The cost is

$$M_{\text{kv}} = (G \cdot T \cdot n_{\text{layers}} \cdot n_{\text{kv\_heads}} \cdot d_{\text{head}} \cdot 2) \cdot \left( \frac{P_{\text{kv}}}{8} + \frac{1}{g_{\text{kv}}} \frac{P_S + P_Z}{8} \right),$$

  where $g_{\text{kv}}$ is the group size, and $P_S$ and $P_Z$ are the precisions of the scales and zero-points. For our experiments, we use symmetric quantization ($P_Z = 0$) with $g_{\text{kv}} = 64$, $P_S = 16$, and test $P_{\text{kv}} \in \{8, 4, 2\}$.

Below are the architectural details and memory footprints for the models used in our experiments (Table 2).

Table 2: **Architectures and memory footprints of evaluated models.**

| Model | Model Size (GB) | | | $n_{\text{layers}}$ | $n_{\text{kv\_heads}}$ | $d_{\text{head}}$ | KV Cache (KB/token) |
|---|---|---|---|---|---|---|---|
| | **4-bit** | **8-bit** | **16-bit** | | | | |
| Qwen3-0.6B | 0.50 | 0.71 | 1.40 | 28 | 8 | 128 | 112 |
| Qwen3-1.7B | 1.26 | 1.93 | 3.78 | 28 | 8 | 128 | 112 |
| Qwen3-4B | 2.49 | 4.19 | 7.49 | 36 | 8 | 128 | 144 |
| Qwen3-8B | 5.68 | 8.94 | 15.26 | 36 | 8 | 128 | 144 |
| Qwen3-14B | 9.30 | 15.50 | 27.51 | 40 | 8 | 128 | 160 |
| Qwen3-32B | 18.01 | 32.66 | 61.02 | 64 | 8 | 128 | 256 |
| R1-Distill/OpenReasoning-1.5B | 1.51 | 2.12 | 3.31 | 28 | 2 | 128 | 28 |
| R1-Distill/OpenReasoning-7B | 5.19 | 8.25 | 14.19 | 28 | 4 | 128 | 56 |
| R1-Distill/OpenReasoning-14B | 9.30 | 15.50 | 27.51 | 48 | 8 | 128 | 192 |

## C  ADDITIONAL EXPERIMENTAL RESULTS

### C.1  LATENCY AND THROUGHPUT ANALYSIS

While we focus primarily on memory–accuracy trade-offs, latency and throughput can be important practical considerations as well. We analyze how model size, weight precision, and generation length affect both metrics.

**Experimental setup.** All measurements are performed on a single NVIDIA A100 80 GB GPU using the vLLM framework (Kwon et al., 2023) with FlashAttention (Dao, 2023) as the attention backend. To measure throughput for a given token budget, we sweep a range of batch sizes and record the highest batch size that completes successfully without out-of-memory errors or KV cache preemption.

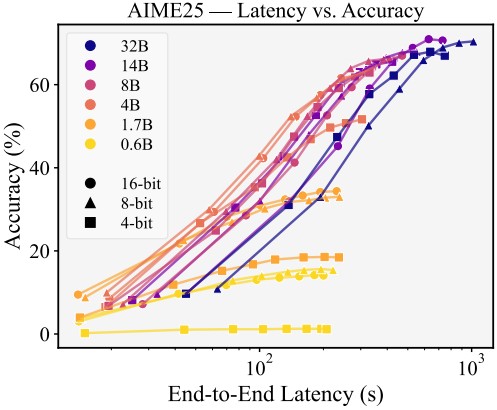

Figure 10: **Latency vs. Accuracy trade-offs (AIME25, Qwen3).** Each curve shows end-to-end latency vs. accuracy for different model sizes and weight precisions with increasing generation length. Generation length emerges as the dominant factor in determining latency, with weight quantization providing more noticeable speedups for large models (14B, 32B).

Figure 11: **Throughput vs. Accuracy trade-offs (AIME25, Qwen3).** Each point represents maximum throughput (requests per second) vs. accuracy under 80 GB VRAM constraints with increasing generation length. While small models can achieve higher batch sizes, the frontier is dominated by configurations that balance model capability with generation efficiency.

We show in Figure 10 that generation length is the dominant factor determining end-to-end latency across all model configurations. The benefit of weight quantization on latency due to reduced memory movement costs is modest for small models (up to 8B) but becomes noticeable for larger models (14B, 32B). For instance, the 14B model takes 137.7 seconds to generate 6k tokens at 16-bit precision, while the 4-bit variant generates 10k tokens in 130.1 seconds. The overall trend for throughput, shown in Figure 11, is similar.

For both latency and throughput, the 4B model with 8-bit and 16-bit precisions consistently demonstrates the strongest speed–accuracy trade-off. Crucially, 4-bit precision is never on the Pareto frontier for any model size, suggesting that for speed-critical applications like reinforcement learning rollouts, higher weight precisions, such as 8-bit, may be the optimal choice (Liu et al., 2025a). The trade-offs become less favorable at the extremes of the scale: small size models (0.6B, 1.7B) achieve extreme batch sizes up to 160 and 170, respectively, yielding throughput of 2.9 and 2.64 requests per second with 2k-token generations, but their accuracy is fundamentally limited. Conversely, the 32B model performs poorly on throughput due to its slow generation speed and large memory footprint, which restricts batching under an 80 GB VRAM constraint.

> **Finding 6**
>
> For both latency and throughput, 4-bit precision is never on the Pareto frontier, as higher precisions (8-bit and 16-bit) consistently provide a better trade-off between accuracy and speed.

## C.2 RESULTS ON OTHER QUANTIZATION METHODS

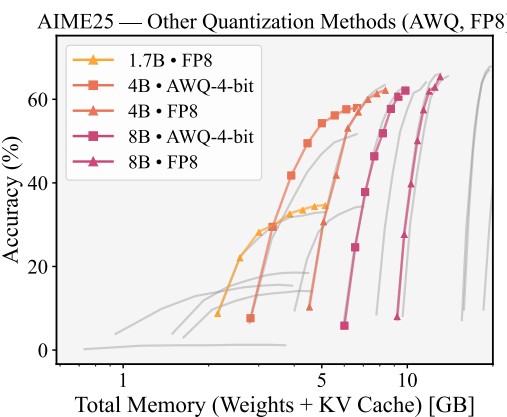

Figure 12: **Comparison of quantization methods.** The background lines correspond to the GPTQ curves from Figure 1. Memory–accuracy trade-offs remain consistent across different quantization schemes.

To test whether our conclusions depend on a particular quantization scheme, we replicated the test-time scaling analysis using AWQ (Lin et al., 2024) and FP8 (Micikevicius et al., 2022) quantization on 1.7B, 4B, and 8B models. As shown in Figure 1, when plotted alongside GPTQ, the memory–accuracy curves from AWQ and FP8 nearly overlap, confirming that the observed trend is not tied to a specific quantization algorithm. Thus, the key memory-allocation pattern persists under these schemes: at a low memory budget (~4.1 GB), a smaller FP8 1.7B model with a long 22k-token generation underperforms a larger AWQ-4-bit 4B model with a shorter 10k-token generation, whereas at a higher budget (~8 GB), an FP8 4B model with 30k tokens outperforms an AWQ-4-bit 8B model with 18k tokens. Overall, the AWQ and FP8 results reinforce our main conclusion that for smaller effective sizes, memory is better spent on model weights, while for larger effective sizes it is better spent on the KV cache to support longer generations.

## C.3 THEORETICAL BATCH SIZE ANALYSIS

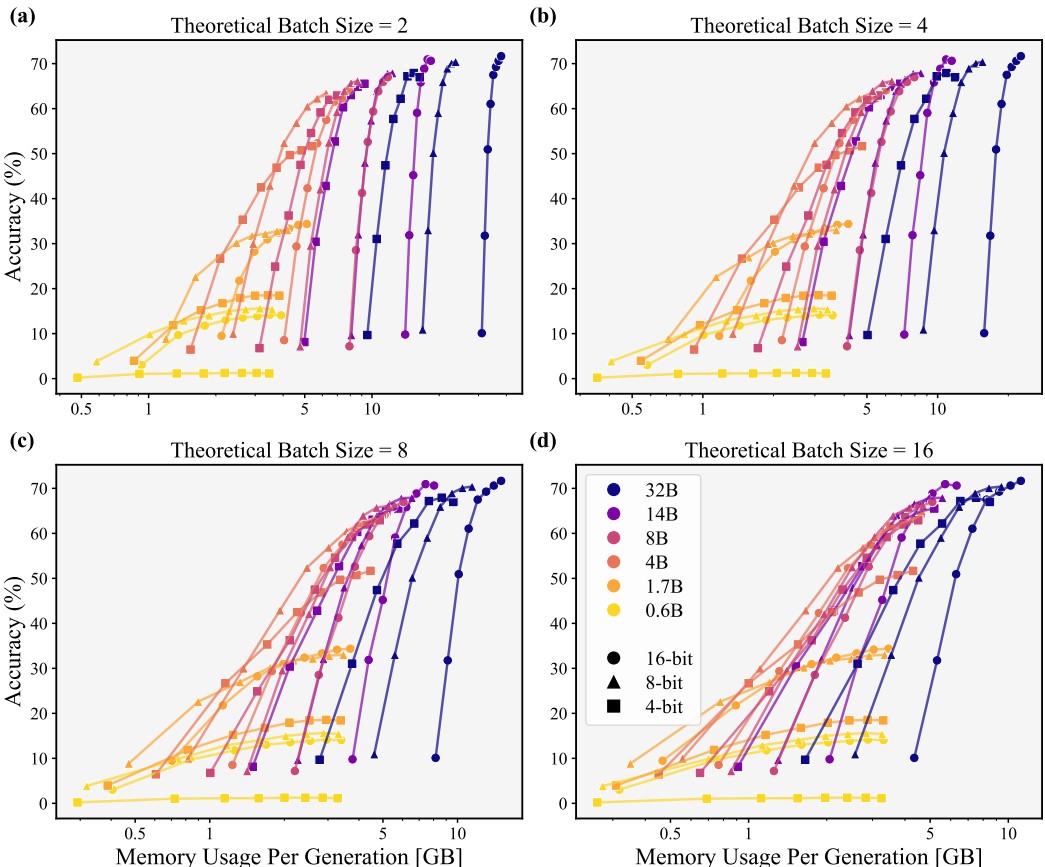

Figure 13: **Memory vs. Accuracy under different theoretical batch sizes (AIME25, Qwen3).** Each subplot shows memory-per-generation vs. accuracy for different theoretical batch sizes, where model weight memory is amortized across concurrent generations. The Pareto frontier shifts as batch size increases, revealing how model weight amortization affects the optimal memory allocation strategy.

Figure 13 examines how memory–accuracy trade-offs change when model weights are shared across multiple concurrent generations, as is common in production serving scenarios. As the theoretical batch size increases, the benefit of smaller model weights diminishes because weight costs are amortized across more generations. We find that the 0.6B model never appears on the Pareto frontier at a theoretical batch size of 16. The 8B and 14B models with 4-bit and 8-bit weight precision and the 4B model with 8-bit and 16-bit precision demonstrate favorable trade-offs in the 1–4 GB memory-per-generation region when the theoretical batch size is 16. Notably, the 8-bit 4B model consistently lies on the Pareto frontier for the 1–2 GB region.

## C.4 DETAILED RESULTS ON LIVECODEBENCH AND MATH500

To provide a more detailed view, we examine LiveCodeBench for code generation and MATH500 for a larger, more diverse set of math problems (Figure 3 and Figure 14).

LiveCodeBench exhibits a trend very similar to AIME25. For models with an effective size below 8-bit 4B, allocating memory to model weights is superior. Under a low memory budget (∼6.2 GB), a 4B 8-bit model with a 14k token budget achieves 61.7% accuracy, significantly outperforming a 1.7B 16-bit model with a larger 22k token budget, which only reaches 42.5%. Conversely, above the threshold, increasing the test-time budget becomes optimal. At a higher budget (∼10.5 GB), a 4B 16-bit model with a 22k token budget reaches 65.00%, whereas a larger 14B 4-bit model with a restricted 6k token budget degrades to 26.8%. Notably, unlike the knowledge-intensive GPQA, 4-bit precision remains less memory-efficient on LiveCodeBench: a 16-bit 4B configuration is strictly more memory-efficient than a 4-bit 14B configuration.

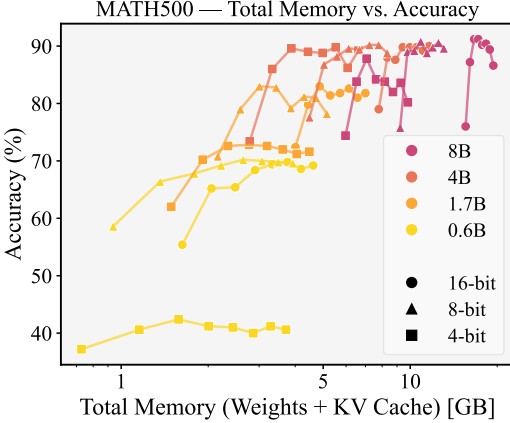

Figure 14: **Memory vs. Accuracy on MATH500 (Qwen3).** Serial scaling on MATH500 also exhibits a scale-dependent memory–accuracy trade-off, but the effective model size threshold shifts toward smaller models due to the benchmark's relative ease.

For MATH500, the easier nature of the task shifts the threshold toward smaller effective sizes. Accuracy saturates quickly once models exceed roughly a 16-bit 1.7B effective size. However, below this threshold, investing in model weights remains more efficient. At a low memory budget (∼3.0 GB), a 1.7B 8-bit model with a 10k token budget achieves 83.0%, outperforming a 0.6B 8-bit model with a 22k token budget at 70.0%. We also observe non-monotonicity in accuracy under budget forcing. This is because for an already-easy set, pushing generation length beyond what is needed can lead to reduced accuracy.

Together, these results reinforce the existence of a scale-dependent threshold and the corresponding memory-optimal strategies. We also find that code generation as well as mathematical reasoning are sensitive to low-bit settings, and higher precision can be more memory-efficient than scaling to a larger model at 4-bit precision under a fixed budget.

## C.5 DETAILED RESULTS FOR PARALLEL SCALING

Figure 15 presents the per-model plots for the parallel scaling analysis discussed in Section 4.

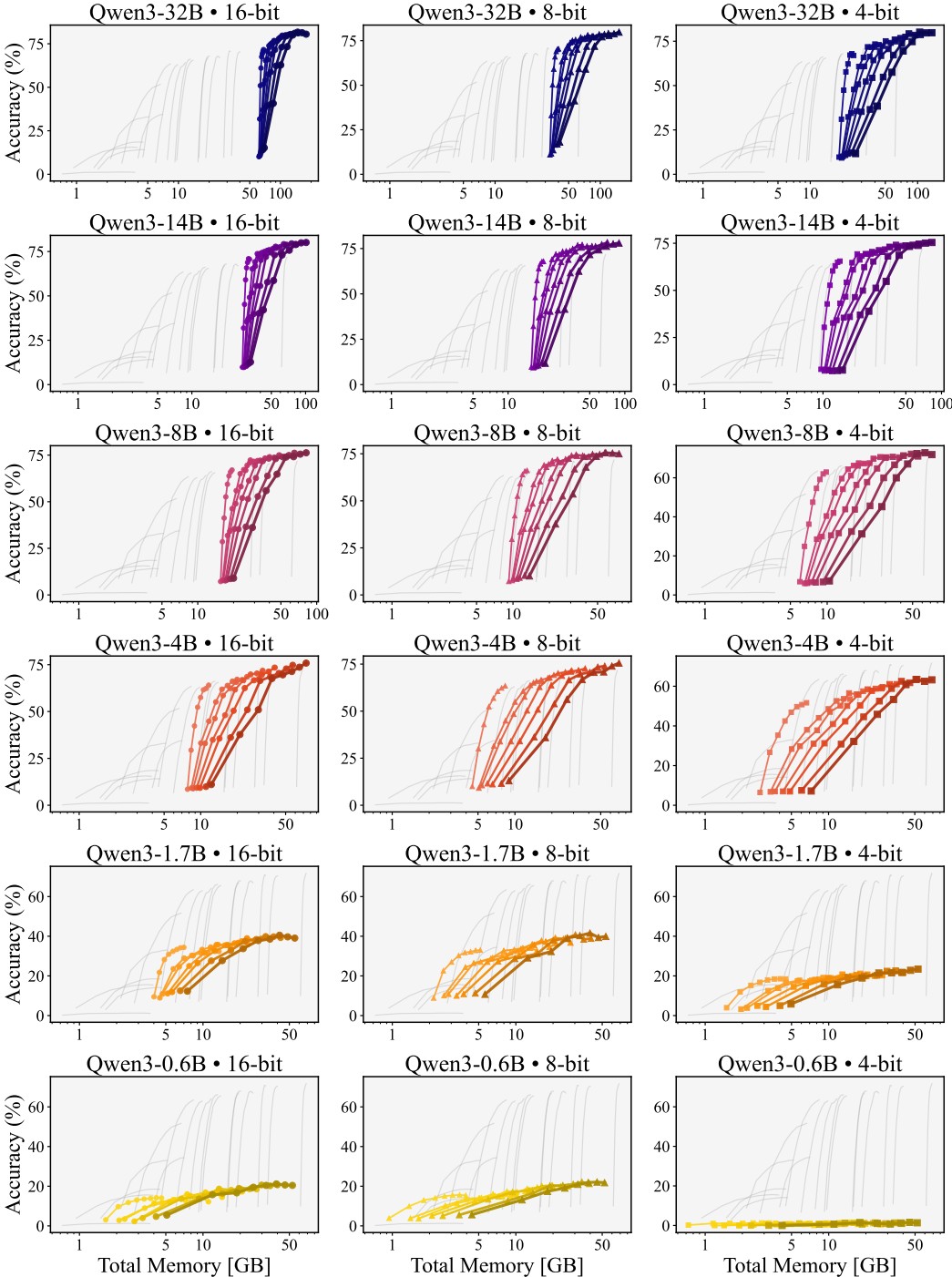

Figure 15: **Per-model Memory vs. Accuracy for parallel scaling (AIME25).** Each plot shows the accuracy-memory trade-off for a single model and weight precision, comparing serial scaling ($G = 1$) with parallel scaling by increasing the sampling group size, $G \in \{1, 3, 4, 6, 8, 12, 16\}$. Points along each curve represent increasing the token budget via budget forcing. Parallel scaling improves the memory–accuracy trade-off, only for models effectively larger than 8-bit 4B.

### C.6 DETAILED RESULTS ON OTHER MODEL FAMILIES

To provide a more detailed view, we examine the DeepSeek-R1-Distill and OpenReasoning-Nemotron reasoning model families (Figures 6 and 16).

For DeepSeek-R1-Distill on AIME25, under a low memory budget ($\sim$9.6 GB), allocating memory to model weights proves superior: a 7B 8-bit model with a 6k token KV cache ($\sim$1.3 GB) achieves 37.09% accuracy, significantly outperforming a 1.5B 8-bit model with a larger 18k token KV cache ($G = 16$, $\sim$7.8 GB) which only reaches 27.60%. Conversely, in a high memory regime ($\sim$30.1 GB), allocating memory to the KV cache becomes optimal: a 7B 16-bit model with 18k tokens and parallel scaling ($G = 16$) reaches 54.5%, surpassing a larger 14B 16-bit model with 14k tokens and serial scaling (39.2%).

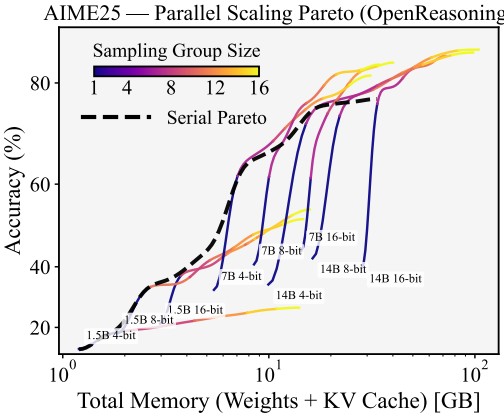

Figure 16: **Parallel scaling for OpenReasoning-Nemotron.** Comparison of serial vs. parallel scaling frontiers. Allocating substantial memory to the KV cache is only effective for sufficiently large models.

Similar trends are observed for OpenReasoning-Nemotron. At a $\sim$6.8 GB budget, the 7B 4-bit model with 26k tokens ($G = 1$) achieves 61.8%, outperforming the 1.5B 16-bit model with 18k tokens and parallel scaling ($G = 8$, 44.4%). At $\sim$31.0 GB, the 7B 16-bit model with 26k tokens and parallel scaling ($G = 12$) reaches 81.1%, surpassing the 14B 16-bit model with 14k tokens (52.5%).

These results confirm that the threshold behavior, favoring model weights for smaller effective sizes and KV cache for larger ones, is consistent across different reasoning model families, although the exact threshold may vary.

## C.7 DETAILED RESULTS FOR KV CACHE COMPRESSION

Figures 17 and 18 show the per-model results for the KV cache compression analysis discussed in Section 5. For eviction, we also present results for StreamingLLM, where we retain the first $T_{\text{retain}}/2$ tokens and the most recent $T_{\text{retain}}/2$ tokens for a given retention budget $T_{\text{retain}}$.

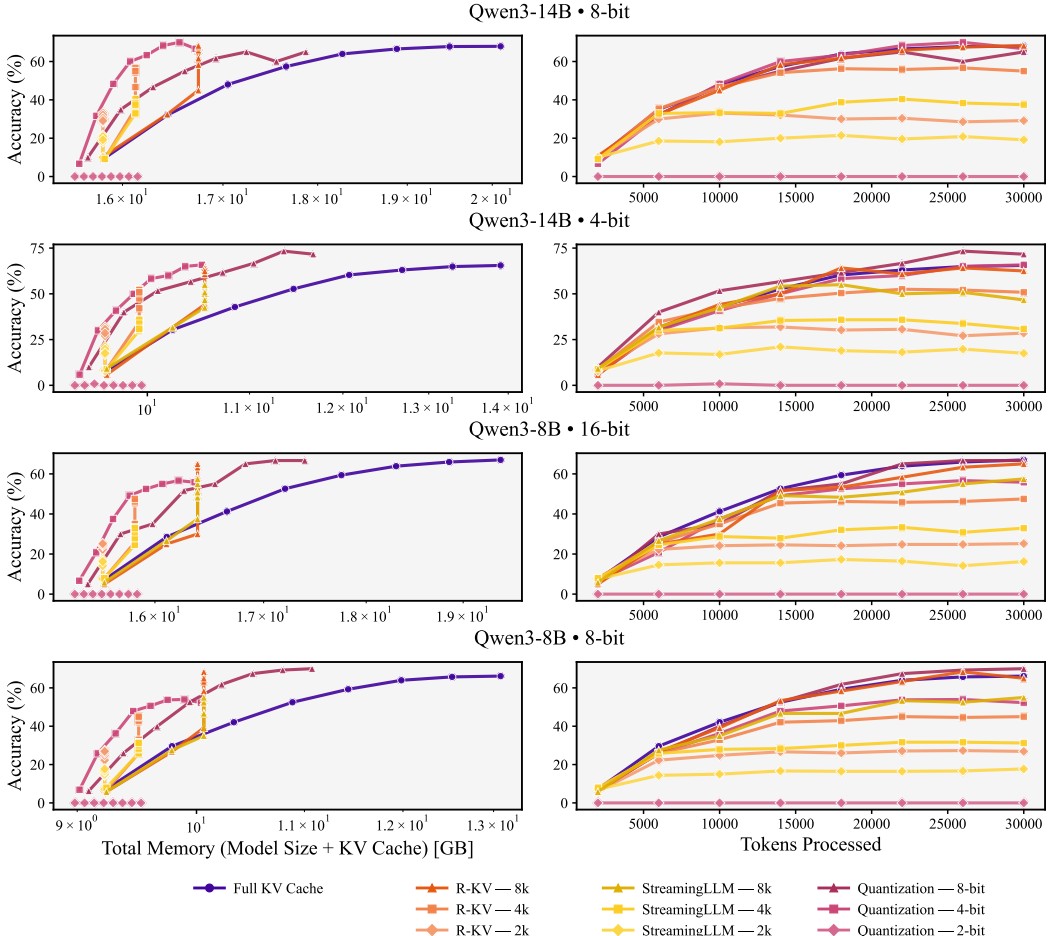

Figure 17: **Per-model Memory vs. Accuracy by KV cache strategy (AIME25, models > 4B).** Each plot shows the accuracy-memory trade-off for a single model and weight precision, comparing KV cache eviction methods (R-KV, StreamingLLM) against KV cache quantization and no compression. Points along each curve represent increasing the number of processed tokens.

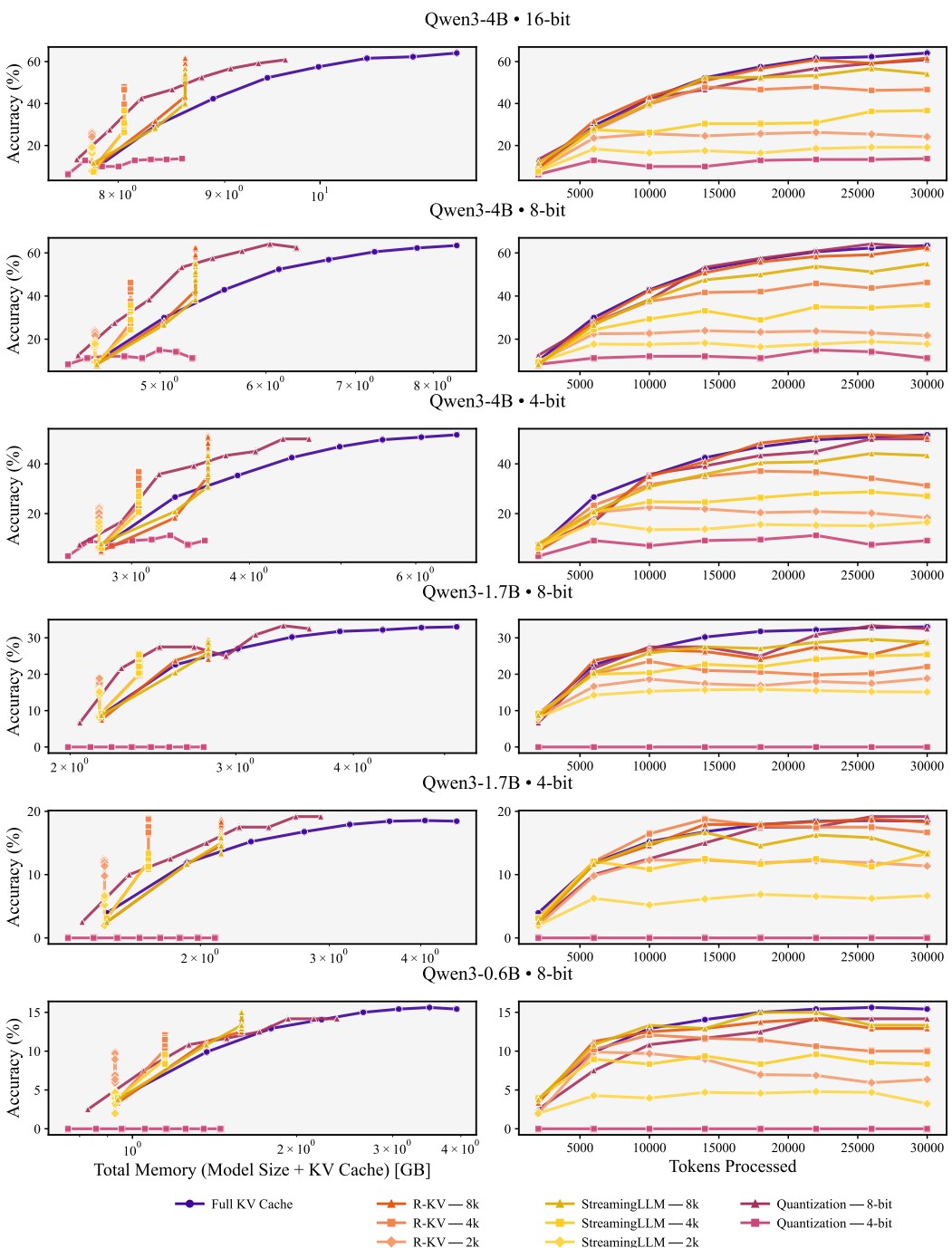

Figure 18: **Per-model Memory vs. Accuracy by KV cache strategy (AIME25, models $\leq$ 4B).** Each plot shows the accuracy-memory trade-off for a single model and weight precision, comparing KV cache eviction methods (R-KV, StreamingLLM) against KV cache quantization and no compression. Points along each curve represent increasing the number of processed tokens.

