# OpenReview forum: "Not All Bits Are Equal: Scale-Dependent Memory Optimization Strategies for Reasoning Models"
_ICLR.cc/2026/Conference — ICLR 2026 Poster_

### Official Review · Reviewer_SZ2z · 2025-10-27

**Soundness:** 2
**Presentation:** 2
**Contribution:** 2
**Rating:** 2
**Confidence:** 4

**Summary:**

The paper studies how to best spend limited memory for reasoning-time compute: more bits for weights vs. longer generations vs. parallel sampling, and how KV-cache strategy (quantize vs. evict) shifts with model scale. Experiments center on Qwen3 variants and two reasoning benchmarks, AIME25 and GPQA-Diamond, claiming a threshold around an “8-bit 4B effective size”, below which weight precision pays more and above which test-time compute (longer or parallel) pays more. The setup is careful inside its sandbox, but it feels like a closed kitchen rather than a street test.

**Strengths:**

The authors execute a clean empirical grid: weight quantization (GPTQ-style), KV compression (HQQ, eviction), token budget, and Best-of-N voting. Within this box, the trends are coherent and useful for practitioners who must fit models under tight VRAM. The memory accounting for weights and KV is neat and transparent, and the two chosen benchmarks are recognizable to the community: AIME-style math and GPQA-Diamond for graduate-level STEM knowledge (AIME25 and GPQA-Diamond are standard touchstones by now).

**Weaknesses:**

For reasoning under fixed memory, there is now a whole ecosystem of **external** baselines that you neither compare nor even acknowledge with experiments. Self-Consistency (SC) is the canonical Best-of-N baseline for chain-of-thought; if you do parallel sampling, you must include SC as a reference decoding policy and report its marginal gains under the same memory budget. Also, **verifier-based** decoding and reranking—PRMs from *Let’s Verify Step by Step* and newer math critics—are exactly what deployment people try before touching KV tricks; ignoring them weakens the claim that “capacity vs. test-time compute” is the main axis. These are not exotic: SC is classic, PRM800K and process reward models are standard, and recent math verifiers show real lift.

On the KV story, the paper pits quantization vs. a couple of eviction settings, but the field moved fast: StreamingLLM (attention-sink), SnapKV, Ada-KV and follow-ups, and fresh 2025–2026 eviction analyses and hybrids. If you argue “evict beats quantize” in certain regimes, please test across these families, otherwise the conclusion feels fragile to method choice. Recent studies also show fragility of eviction heuristics across layers/heads/tasks; you must address that, or at least show sensitivity analyses.

The evaluation set is too narrow. AIME25 and GPQA-Diamond are fine sanity checks, but the claim is about *reasoning deployment under memory constraints*. Put your recipe into the streets: long-context multi-doc QA (LongBench/v2), multi-hop (HotpotQA), program synthesis/real repos (SWE-bench), and cross-domain text-to-SQL (Spider). If your prescriptions survive these, the paper's claim will be more valid.

**Questions:**

If you add SC and PRM-reranked Best-of-N under the same VRAM cap, do your conclusions still hold? On KV methods, do SnapKV/Ada-KV/StreamingLLM change the “evict beats quantize for small models” rule, or only for certain context mixes? Finally, can you show one real-world slice, say, LongBench multi-doc QA and SWE-bench Verified Mini, where your policy improves both accuracy and cost per solved issue compared to SC or PRM-reranked decoding?

---

> ### Author Response · Authors · 2025-11-21
> **Authors' Response to Reviewer SZ2z**
>
> We thank the reviewer for their valuable feedback. We are pleased to hear that you found our experiments careful and transparent. Other reviewers (tfiR, EX7S, Daoc) have also acknowledged the importance of our research question and its significance for practitioners, and identified our rigorous and systematic experimental design as a strength. We clarify and address your concerns below and would be grateful if you would consider revisiting your evaluation after reviewing our responses.
>
> **W1. The absence of self-consistency (SC) as a baseline**
>
> The reviewer suggests that we did not include or acknowledge self-consistency as a baseline. This is not true. We would like to clarify that **self-consistency is already implemented in our parallel sampling analysis**. Specifically, we generate multiple independent reasoning trajectories at temperature 0.6 and select the final answer using majority voting, which corresponds to the canonical self-consistency aggregation method proposed by the original SC paper [1]. In fact, we explicitly describe this process and cite Wang et al. in the initial submission of our paper:
> > L160–161: "In its simplest form, without any external model, majority voting selects the final answer as the most frequent among the independently sampled outputs (Wang et al., 2022)."
> > L317–319: "For systematic evaluation, we use budget forcing to control the token budget for each of the G parallel samples and use majority voting to select the final answer."
>
> Although the SC paper introduces several aggregation strategies, including normalized probability-based voting and simple majority voting, majority voting is the most commonly adopted form of SC, as also shown in [2, 3]. This is because Figure 1 of [1] shows that majority voting achieves equal or better accuracy than other methods, which justifies our choice to adopt it for our parallel scaling experiments.
>
> To ensure that this connection is clearer to readers, we append a description in the revised draft that our "majority voting" corresponds to **self-consistency with majority aggregation**. (L319-320)
>
> **W2&Q1. Examines KV cache compression, but lacks verifier-based decoding and PRMs**
>
> We did not include methods that require external models because they incur substantial additional memory overhead relative to their marginal accuracy gains, leading to a poor memory–accuracy trade-off. KV cache compression techniques aim to improve the deployment-time memory trade-off, while verifiers and reward models are not designed for deployment-time memory efficiency. Hence, it is unclear to us whether the claim that “verifier-based decoding and reranking—PRMs—are exactly what deployment people try before touching KV tricks” applies to memory-constrained deployment settings.
>
> > If you add PRM-reranked Best-of-N under the same VRAM cap, do your conclusions still hold?
>
> Yes, our conclusion still holds, and **using a PRM is strictly worse** than parallel scaling with majority voting. To support this point, we additionally evaluate **Best-of-N parallel scaling with ActPRM-X** [4], the state-of-the-art model on ProcessBench and PRMBench. Following [4], we use the model to produce step-level scores for each of the N independent trajectories and aggregate them by product to obtain a single score per trajectory. We then pick the candidate with the highest PRM score and count success if that candidate is ground-truth correct. We ran this evaluation across the Qwen3 model family with 4/8/16-bit precision on AIME25. For the total memory calculation, we include the additional weight of the 7B ActPRM-X model (13.28 GB) as a constant factor.
>
> In Figure 18 of the revised draft, we compare the three Pareto frontiers formed by serial scaling, parallel scaling with majority voting, and parallel scaling with Best-of-N using PRM. We find that even with the current best-performing PRM, the accuracy gains obtained by using more GPU memory for the external model are marginal, leading to a **worse memory–accuracy frontier than majority voting**; in the low-memory regime, it is even less favorable than serial scaling alone due to the PRM’s size.
>
> [1] Wang, Xuezhi, et al. "Self-consistency improves chain of thought reasoning in language models." ICLR, 2023
>
> [2] Touvron, Hugo, et al. "LLaMA: Open and Efficient Foundation Language Models." ArXiv, 2023
>
> [3] Yao, Shunyu, et al. "Tree of thoughts: Deliberate problem solving with large language models." NeurIPS, 2023
>
> [4] Duan, Keyu, et al. "Efficient process reward model training via active learning." ArXiv, 2025

---

> > ### Author Response · Authors · 2025-11-21
> > **Authors' Response to Reviewer SZ2z**
> >
> > **W3&Q2. On KV cache eviction methods**
> >
> > > The paper pits quantization vs. a couple of eviction settings, but the field moved fast: StreamingLLM (attention-sink), SnapKV, Ada-KV and follow-ups, and fresh 2025–2026 eviction analyses and hybrids.
> >
> > We clarify that **the methods suggested by the reviewer, SnapKV and Ada-KV, perform prefill-only eviction**, meaning they compress the KV cache from the input context but do **not** operate during decoding. The lack of decoding-time eviction is crucial in our selection of KV eviction methods, since we are examining large reasoning models that have longer outputs than inputs. The only method among those suggested that supports decoding-time eviction is StreamingLLM, for which we already provide results in Appendix C.4 of our submitted manuscript.
> >
> > If the reviewer is suggesting testing the prefill-only methods by applying them iteratively during decoding, we note that **R-KV** [5], which we employ in our work, is built upon these attention-based approaches and introduces redundancy-aware improvements specifically tailored for reasoning models. The R-KV paper demonstrates strict superiority over iterated SnapKV.
> >
> > Among the two eviction methods that we tested, StreamingLLM and R-KV, we view StreamingLLM as a lower bound for decoding-time eviction algorithms, as it does not use attention-pattern information. In contrast, R-KV serves as a strong upper bound of eviction methods. Together, these two methods cover a broad spectrum of the current landscape for decoding-time eviction algorithms.
> >
> > Regarding recency, R-KV is among the most recently introduced eviction methods (May 2025), compared with the suggested StreamingLLM (Sep 2023), SnapKV (Apr 2024), and Ada-KV (Jul 2024). Thus, evaluating R-KV and StreamingLLM provides a comprehensive and up-to-date comparison.
> >
> > > On KV methods, do SnapKV/Ada-KV/StreamingLLM change the “evict beats quantize for small models” rule, or only for certain context mixes?
> >
> > In Figure 12 of our initial submission, StreamingLLM matched or exceeded the memory efficiency of 8-bit KV cache quantization for small models, supporting our main finding that **eviction is more memory-efficient than quantization in this regime**.
> >
> > [5] Cai, Zefan, et al. "R-KV: Redundancy-aware KV Cache Compression for Training-Free Reasoning Models Acceleration." ArXiv, 2025

---

> > > ### Author Response · Authors · 2025-11-21
> > > **Authors' Response to Reviewer SZ2z**
> > >
> > > **W4&Q3. On task diversity and narrow evaluation scope**
> > >
> > > We first clarify that our study *intentionally* focuses on long CoT reasoning tasks. Some of the suggested benchmarks, such as LongBench and HotPotQA, primarily evaluate **long-context retrieval with short generations**. In contrast, our analysis targets long-generation scenarios, where the KV cache becomes the dominant memory factor. Consequently, the memory–accuracy trade-offs and optimal scaling policies in these two settings differ. Prior work [6,7,8] has **already established memory-optimal prescriptions for short-generation tasks**. Concretely, they recommend aggressive weight quantization near and below 4-bit when the KV cache is negligible. Our results complement these by extending the analysis to the long-generation regime, where the memory allocation between model weights and the KV cache plays a central role.
> > >
> > > As suggested by the reviewer, to broaden the evaluation within long reasoning, we include two additional benchmarks: **LiveCodeBench** for reasoning in code generation and **MATH500** for a larger and more diverse set of math questions. Please see Figures 15 and 16 of the revised manuscript.
> > >
> > > For LiveCodeBench, we find a **trend very similar to AIME25**: for models with effective size below 8-bit 4B, allocating more memory to model weights yields larger gains, whereas above this threshold, memory is better spent increasing the test-time budget until performance saturates. We also show this in the table below, where we compare, within matched memory budgets, pairs of configurations that allocate **more memory to the KV cache** (upper row) versus **more memory to model weights** (lower row) in both low- and high-memory regimes. Also, unlike knowledge-intensive GPQA, **4-bit precision remains less memory-efficient** on LiveCodeBench: a 16-bit 4B configuration is strictly more memory-efficient than a 4-bit 14B configuration.
> > >
> > > ***LiveCodeBench (Qwen3)***
> > > |**Low-memory budget** (~6.2 GB)||**Acc. (%)**|**High-memory budget** (~10.5 GB)||**Acc. (%)**|
> > > |-|-|-:|-|-|-:|
> > > |**Model weights**|**KV cache**||**Model weights**|**KV cache**||
> > > |1.7B · 16-bit (3.78 GB)|22k tokens (2.43 GB)|42.50|4B · 16-bit (7.49 GB)|22k tokens (3.13 GB)|**65.00**|
> > > |4B · 8-bit (4.19 GB)|14k tokens (2.03 GB)|**61.69**|14B · 4-bit (9.30 GB)|6k tokens (1.03 GB)|26.75|
> > >
> > > For MATH500, the **threshold shifts left** (toward smaller effective sizes) because the benchmark is easier than AIME25. Accuracy saturates quickly once models exceed roughly a 16-bit 1.7B effective size. However, for models below this memory threshold, investing memory in model weights is still more efficient than growing the KV cache, as shown in the table below. We also observe non-monotonicity in accuracy under budget forcing: for an already easy set, pushing generation length beyond what is needed can lead to reduced accuracy.
> > >
> > > ***MATH500 (Qwen3)***
> > > |**Low-memory budget** (~3.0 GB)||**Acc. (%)**|**High-memory budget** (~7.2 GB)||**Acc. (%)**|
> > > |-|-|-:|-|-|-:|
> > > |**Model weights**|**KV cache**||**Model weights**|**KV cache**||
> > > |0.6B · 8-bit (0.71 GB)|22k tokens (2.36 GB)|70.0|4B · 8-bit (4.19 GB)|22k tokens (3.04 GB)|**90.2**|
> > > |1.7B · 8-bit (1.93 GB)|10k tokens (1.08 GB)|**83.0**|8B · 4-bit (5.68 GB)|10k tokens (1.39 GB)|87.8|
> > >
> > > Together, these results reinforce the **existence of a scale-dependent threshold** and the **corresponding memory-optimal strategies** on either side of it. They also strengthen our claim about numerical precision: reasoning in math and code is sensitive to low-bit settings, and higher precision can be more memory-efficient than scaling to a larger model at 4-bit precision under a fixed budget.
> > >
> > > [6] Dettmers, Tim, and Luke Zettlemoyer. "The case for 4-bit precision: k-bit inference scaling laws." ICML, 2023
> > >
> > > [7] Frantar, Elias, et al. "Gptq: Accurate post-training quantization for generative pre-trained transformers." ICLR, 2023
> > >
> > > [8] Chee, Jerry, et al. "Quip: 2-bit quantization of large language models with guarantees." NeurIPS, 2023

---

> > > > ### Author Response · Authors · 2025-11-28
> > > >
> > > > We thank you again for the questions you raised in your initial review.
> > > >
> > > > During the rebuttal phase, we have put substantial effort into clarifying the points you highlighted and running additional experiments to address them.
> > > > These additions have helped address similar concerns raised by other reviewers, who have responded positively to the revised evidence and clarifications.
> > > >
> > > > As the rebuttal period is drawing to a close, we would be very grateful if you could briefly revisit our response and consider whether any of your earlier concerns have been resolved.

---

### Official Review · Reviewer_Daoc · 2025-11-01

**Soundness:** 3
**Presentation:** 3
**Contribution:** 3
**Rating:** 6
**Confidence:** 3

**Summary:**

This paper investigates memory-optimal inference strategies for reasoning models under fixed memory budgets. Through systematic experiments on the Qwen3 model family (0.6B to 32B parameters) across mathematical and knowledge-intensive reasoning tasks, the authors explore trade-offs between model size, weight precision, generation length, parallel scaling, and KV cache compression. The key finding is that optimal strategies are scale-dependent rather than universal: models effectively smaller than 8-bit 4B parameters achieve better accuracy by allocating memory to larger model weights rather than extended generation, while larger models benefit from maximizing test-time compute. Additionally, mathematical reasoning tasks require higher weight precision (8/16-bit) compared to knowledge-intensive tasks where 4-bit quantization is optimal. The work challenges the conventional wisdom that 4-bit quantization is universally memory-optimal and provides principled guidelines for practitioners deploying reasoning models.

**Strengths:**

The paper's main strength is its comprehensive and rigorous empirical investigation of a practically important problem. The systematic exploration of over 1,700 configurations across multiple optimization dimensions provides valuable insights that challenge existing practices. The finding that memory-optimal strategies are fundamentally scale-dependent—with the "8-bit 4B" threshold determining whether to prioritize model capacity or test-time compute—is significant for practitioners. The discovery that task characteristics matter (mathematical reasoning requiring higher precision than knowledge-intensive tasks) suggests important considerations for task-specific deployment. The experimental methodology is thorough with clearly defined memory equations, and the presentation using Pareto frontier analysis effectively communicates complex trade-offs. The paper addresses a critical practical challenge as reasoning models with extended generation make KV cache memory a dominant factor, requiring rethinking of compression strategies.

**Weaknesses:**

The primary limitation is the narrow evaluation scope—findings are based solely on the Qwen3 model family and two benchmarks (AIME25 and GPQA-Diamond), raising significant questions about generalizability to other architectures and reasoning domains. The paper identifies "8-bit 4B" as a critical threshold across multiple findings but provides no theoretical justification for why this specific scale matters, making it unclear whether this threshold would transfer to other model families or evolve as architectures change. The analysis is primarily observational, documenting empirical trade-offs without providing mechanistic understanding of why mathematical reasoning is more sensitive to quantization or why the inflection point occurs at this particular scale. The exclusion of methods using external models (verifiers, process reward models) limits practical applicability, as these are commonly used in production. While Appendix C.2 addresses batched inference scenarios, the main analysis assumes single-inference settings which may not reflect typical deployment constraints where model weights are amortized across requests.

**Questions:**

The "8-bit 4B" threshold emerges as critical across multiple findings. Can you provide insight into why this specific effective size serves as the inflection point? Is there something fundamental about this capacity level, or could it be an artifact of the Qwen3 architecture? Have you explored whether this threshold shifts for other model families or architectural choices?

---

> ### Author Response · Authors · 2025-11-21
> **Authors' Response to Reviewer Daoc**
>
> Thank you for the thoughtful review. We are glad the reviewer highlighted the **rigorous, comprehensive experimentation** over 1,700 configurations and the **practical significance** of our findings for practitioners. These first two points were also echoed by the other reviewers (tfiR, EX7S). We also appreciate your comments on the clear communication of the findings, which aligns with feedback from Reviewer tfiR.
>
> **W1. Findings are solely based on Qwen3 family**
>
> This is valid feedback and aligns with one of the limitations we acknowledged in Section 7. Conducting comparable analyses across many model families is challenging because few reasoning model families currently provide a fine-grained scale ladder like Qwen3.
>
> Nonetheless, we have conducted additional experiments on two other model families: **DeepSeek-R1-Distill** (1.5B–14B) and **OpenReasoning-Nemotron** (1.5B–14B) on AIME25.
>
> ***OpenReasoning-Nemotron***
> |**Low-memory budget** (~6.8 GB)||**Acc. (%)**|**High-memory budget** (~31.0 GB)||**Acc. (%)**|
> |-|-|-:|-|-|-:|
> |**Model weights**|**KV cache**||**Model weights**|**KV cache**||
> |1.5B · 16-bit (2.91 GB)|18k tokens, g=8 (3.88 GB)|44.38|7B · 16-bit (14.27 GB)|26k tokens, g=12 (16.76 GB)|**81.14**|
> |7B · 4-bit (5.30 GB)|26k tokens, g=1 (1.40 GB)|**61.77**|14B · 16-bit (27.90 GB)|14k tokens, g=1 (2.59 GB)|52.50|
>
> ***DeepSeek-R1-Distill***
> |**Low-memory budget** (~9.6 GB)||**Acc. (%)**|**High-memory budget** (~30.1 GB)||**Acc. (%)**|
> |-|-|-:|-|-|-:|
> |**Model weights**|**KV cache**||**Model weights**|**KV cache**||
> |1.5B · 8-bit (2.12 GB)|18k tokens, g=16 (7.76 GB)|27.60|7B · 16-bit (14.19 GB)|18k tokens, g=16 (15.52 GB)|**54.52**|
> |7B · 8-bit (8.25 GB)|6k tokens, g=4 (1.31 GB)|**37.09**|14B · 16-bit (27.51 GB)|14k tokens, g=1 (2.59 GB)|39.17|
>
> In these tables, we compare, within matched memory budgets, pairs of configurations that allocate **more memory to the KV cache** (upper row) versus **more memory to model weights** (lower row) in both low- and high-memory regimes. Across both families, our main conclusion holds:
> > For smaller models, it is more memory-efficient to allocate memory to model weights; for larger models, to allocate memory to the KV cache for longer generations (L258).
>
> We also show the full landscape in **Figures 13 and 14** of the revised manuscript. For R1-Distill, parallel scaling of the 4-bit 7B model performs slightly below the Pareto frontier of serial scaling. For OpenReasoning-Nemotron, the 4-bit 7B model achieves a marginally stronger memory–accuracy trade-off. For both model families, models effectively smaller than this threshold show suboptimal memory trade-offs when forced to spend memory on the KV cache, while models effectively larger than the threshold benefit from parallel scaling. While the exact memory threshold shifts slightly by model family, the **existence of such a threshold** and the resulting memory-optimal scaling strategies remain *consistent* across reasoning model families.

---

> > ### Author Response · Authors · 2025-11-21
> > **Authors' Response to Reviewer Daoc**
> >
> > **W2. Findings are solely based on two benchmarks**
> >
> > We agree that extending the evaluation set beyond AIME25 and GPQA-Diamond enhances the robustness of our conclusions. Accordingly, we have conducted experiments on two additional reasoning benchmarks: **LiveCodeBench** for reasoning in code generation and **MATH500** for a larger and more diverse set of math questions. Please refer to **Figures 15 and 16** of the revised manuscript.
> >
> > For LiveCodeBench, we find a **trend very similar to AIME25**: for models with effective size below 8-bit 4B, allocating more memory to model weights yields larger gains, whereas above this threshold, memory is better spent increasing the test-time budget until performance saturates. We also show this in the table below, where we compare, within matched memory budgets, configurations that allocate **more memory to the KV cache** (upper row) versus **more memory to model weights** (lower row) in both low- and high-memory regimes. Notably, unlike knowledge-intensive GPQA, **4-bit precision remains less memory-efficient** on LiveCodeBench: a 16-bit 4B configuration is strictly more memory-efficient than a 4-bit 14B configuration.
> >
> > ***LiveCodeBench (Qwen3)***
> > |**Low-memory budget** (~6.2 GB)||**Acc. (%)**|**High-memory budget** (~10.5 GB)||**Acc. (%)**|
> > |-|-|-:|-|-|-:|
> > |**Model weights**|**KV cache**||**Model weights**|**KV cache**||
> > |1.7B · 16-bit (3.78 GB)|22k tokens (2.43 GB)|42.50|4B · 16-bit (7.49 GB)|22k tokens (3.13 GB)|**65.00**|
> > |4B · 8-bit (4.19 GB)|14k tokens (2.03 GB)|**61.69**|14B · 4-bit (9.30 GB)|6k tokens (1.03 GB)|26.75|
> >
> > For MATH500, the **threshold shifts left** (toward smaller effective sizes) because the benchmark is easier than AIME25. Accuracy saturates quickly once models exceed roughly a 16-bit 1.7B effective size. However, for models below this memory threshold, investing memory in model weights is still more efficient than growing the KV cache, as shown in the table below. We also observe non-monotonicity in accuracy under budget forcing: for an already easy set, pushing generation length beyond what is needed can reduce accuracy.
> >
> > ***MATH500 (Qwen3)***
> > |**Low-memory budget** (~3.0 GB)||**Acc. (%)**|**High-memory budget** (~7.2 GB)||**Acc. (%)**|
> > |-|-|-:|-|-|-:|
> > |**Model weights**|**KV cache**||**Model weights**|**KV cache**||
> > |0.6B · 8-bit (0.71 GB)|22k tokens (2.36 GB)|70.0|4B · 8-bit (4.19 GB)|22k tokens (3.04 GB)|**90.2**|
> > |1.7B · 8-bit (1.93 GB)|10k tokens (1.08 GB)|**83.0**|8B · 4-bit (5.68 GB)|10k tokens (1.39 GB)|87.8|
> >
> > Together, these results reinforce the **existence of a scale-dependent threshold** and the **corresponding memory-optimal strategies** on either side of it. They also strengthen our claim about numerical precision: reasoning in math and code is sensitive to low-bit settings, and higher precision can be more memory-efficient than scaling to a larger model at 4-bit precision under a fixed budget.

---

> > > ### Author Response · Authors · 2025-11-21
> > > **Authors' Response to Reviewer Daoc**
> > >
> > > **W3&Q1. On the memory threshold and theoretical justification**
> > >
> > > We appreciate this thoughtful comment. The identification of the memory threshold for determining memory-optimal reasoning is indeed an empirical finding, and we view this as a key contribution of the paper rather than a limitation. To our knowledge, this is the first study to reveal a scale-dependent transition in memory-optimal reasoning. As in many prior studies on scaling behavior [1–4], our goal is to establish empirical findings that can guide deployment strategies.
> > >
> > > To test whether this scale-dependent behavior generalizes, we extended our analysis to two additional model families (DeepSeek-R1-Distill and OpenReasoning-Nemotron) and two additional tasks (LiveCodeBench and MATH500), as mentioned for W1&2. Across all cases, the exact memory threshold shifts slightly with model family and task, but the main conclusion remains consistent:
> > > > We find that the memory-optimal inference strategy for reasoning models *cannot be a one-size-fits-all prescription*: instead, it depends on the model’s capacity (determined by effective size) and the nature of the task. (L462–464)
> > >
> > > We believe that identifying this scale-dependent behavior is itself a novel and practically valuable finding even without a theoretical explanation. Moreover, the particular inflection point of 8-bit 4B itself is not special. As we state in L466–468, "the inflection point where extra KV cache beats extra model weight may change as models become more sophisticated." Thus, no, there is nothing fundamental about this capacity level (it is rather indicative of model capabilities in 2025), nor is it simply an artifact of Qwen3 (we observe similar thresholds for other model families). However, we provide strong evidence that the existence of such a threshold is fundamental to test-time scaling if practitioners are to be memory-efficient.
> > >
> > > **W4. Absence of methods using external models**
> > >
> > > We disagree that the "exclusion of methods using external models (verifiers, process reward models) limits practical applicability." While external verifier-based or process reward model (PRM) decoding methods are used to improve reasoning accuracy, these methods are not necessarily designed for memory-efficient inference and introduce substantial memory overhead, as they require a separate model (often several billion parameters). Since our work specifically targets memory-optimal reasoning strategies, we did not include these methods in our main analysis.
> > >
> > > Our conclusion still holds, and **using a PRM is strictly worse than** parallel scaling with majority voting. To substantiate this point, we additionally evaluate **Best-of-N parallel scaling with ActPRM-X** [5], the state-of-the-art model on ProcessBench and PRMBench. Following [5], we use the model to produce step-level scores for each of the N independent trajectories and aggregate them by product to obtain a single score per trajectory. We then pick the candidate with the highest PRM score and count success if that candidate is ground-truth correct. We ran this evaluation across the Qwen3 model family with 4/8/16-bit precision on AIME25. For the total memory calculation, we include the additional weight of the 7B ActPRM-X model (13.28 GB) as a constant factor.
> > >
> > > In Figure 18 of the revised draft, we compare the three Pareto frontiers formed by serial scaling, parallel scaling via majority voting, and parallel scaling with Best-of-N using PRM. We find that even with the current best-performing PRM, the accuracy gains obtained by trading GPU memory for the external model are marginal, leading to a **much worse memory–accuracy frontier than majority voting**; in the low-memory regime, it is even less favorable than serial scaling alone due to the PRM’s relatively large size. Even considering a multi-batch inference setting where both the LLM and PRM model weights are amortized, we see that the use of a PRM does not provide much better memory efficiency than parallel scaling with majority voting due to its marginal gain in accuracy.
> > >
> > > [1] Dettmers, Tim, and Luke Zettlemoyer. "The case for 4-bit precision: k-bit inference scaling laws." ICML, 2023
> > >
> > > [2] Kumar, Tanishq, et al. "Scaling Laws for Precision." ICLR, 2025
> > >
> > > [3] Kaplan, Jared, et al. "Scaling laws for neural language models." ArXiv, 2020
> > >
> > > [4] Hoffmann, Jordan, et al. "Training compute-optimal large language models." NeurIPS, 2022
> > >
> > > [5] Duan, Keyu, et al. "Efficient process reward model training via active learning." ArXiv, 2025

---

> > > > ### Author Response · Authors · 2025-11-21
> > > > **Authors' Response to Reviewer Daoc**
> > > >
> > > > **W5. Main analysis assumes single-inference settings**
> > > >
> > > > > The main analysis assumes single-inference settings, which may not reflect typical deployment constraints where model weights are amortized across requests.
> > > >
> > > > We thank the reviewer for their valuable feedback. However, we respectfully disagree with the comment for three reasons:
> > > >
> > > > 1. Not only does Appendix C.2 provide insight into the batched inference setting, but many of our main findings also apply to batched inference. For instance, our takeaways from KV cache compression (Findings 4 and 5) remain true regardless of the setting.
> > > > 2. Our main analysis assumes a single-inference scenario for clarity of comparison, but our findings based on single-inference remain highly relevant for *on-device* deployments. On-device language model usage now represents a significant and growing share of real-world inference scenarios, as seen in [6–9] and references therein. These applications typically operate with a **single batch size**, face **strict memory limits**, and target a **specific task domain**. Such settings directly align with our main analysis.
> > > > 3. Prior work does *not* systematically study even the single-inference setting. As the reviewer has correctly noted, multi-batch inference amortizes model-weight costs across requests, further amplifying the significance of KV cache memory and pushing deployments into the KV-dominated regime that motivates our study. In contrast, prior works [10–12] on non-reasoning or short-generation tasks operate in the regime of negligible KV memory, where optimization largely concerns model weights.
> > > >
> > > > [6] Saad-Falcon, Jon, et al. "Intelligence per Watt: Measuring Intelligence Efficiency of Local AI." ArXiv, 2025
> > > >
> > > > [7] Qu, Guanqiao, et al. "Mobile edge intelligence for large language models: A contemporary survey." IEEE Communications Surveys & Tutorials, 2025
> > > >
> > > > [8] Zheng, Yue, et al. "A review on edge large language models: Design, execution, and applications." ACM Computing Surveys, 2025
> > > >
> > > > [9] Liu, Zechun, et al. "Mobilellm: Optimizing sub-billion parameter language models for on-device use cases." ICML, 2024
> > > >
> > > > [10] Dettmers, Tim, and Luke Zettlemoyer. "The case for 4-bit precision: k-bit inference scaling laws." ICML, 2023
> > > >
> > > > [11] Frantar, Elias, et al. "Gptq: Accurate post-training quantization for generative pre-trained transformers." ICLR, 2023
> > > >
> > > > [12] Chee, Jerry, et al. "Quip: 2-bit quantization of large language models with guarantees." NeurIPS, 2023

---

> > > > > ### Comment · Reviewer_Daoc · 2025-11-26
> > > > >
> > > > > My concerns are addressed. I will raise my score.

---

> > > > > > ### Author Response · Authors · 2025-11-28
> > > > > >
> > > > > > We are very glad that our rebuttal successfully addressed your concerns, and we sincerely appreciate your decision to raise the score.
> > > > > > Thank you as well for your questions and suggestions, they will undoubtedly help us further improve the quality of the paper.

---

### Official Review · Reviewer_EX7S · 2025-11-01

**Soundness:** 3
**Presentation:** 4
**Contribution:** 3
**Rating:** 6
**Confidence:** 3

**Summary:**

This paper investigates memory-optimal inference strategies for LLMs on reasoning tasks, operating under a fixed total memory budget. The authors challenge the prevailing consensus that 4-bit quantization is universally optimal, demonstrating that for long-generation reasoning tasks, the KV cache can become the dominant memory bottleneck. Through a systematic empirical study (over 1,700 configurations) on the Qwen3 model family, the paper analyzes the trade-offs between model size, weight precision, parallel scaling, and KV cache compression. The paper provide several key findings on the trade-offs and propose some guideline to achieve better memory efficiency.

**Strengths:**

1. Exhaustive Experimentation: The study is commendable for its comprehensive empirical approach, evaluating a wide range of model sizes (0.6B to 32B) and systematically comparing various KV cache optimization methods against a baseline.

2. Focus on a Practical Metric: The paper correctly centers its analysis on accuracy as the core metric for evaluating inference efficiency under memory constraints. This approach has direct and significant value for real-world deployment scenarios.

3. Systematic, Multi-Dimensional Analysis: The work provides a strong, multi-dimensional analysis of the complex trade-offs involved in model deployment. The resulting findings offer principled guidelines and empirical rules that can genuinely inform practical decisions in model configuration.

**Weaknesses:**

1. Choice of AIME25 Benchmark: The AIME25 dataset may not be a suitable benchmark for drawing generalizable conclusions. Its small size (only 30 problems) can lead to high variance in results, and its extreme difficulty is heavily reliant on a model's peak mathematical reasoning capability. The small performance differences observed may not be sufficient evidence of a decisive performance gap.

2. Model-Specific Findings: As the authors acknowledge in the limitations, the study is confined exclusively to the Qwen3 model family. This focus risks overfitting the conclusions to the specific capabilities and performance characteristics of the Qwen3 architecture.

3. Limited Task Diversity: The selected benchmarks (AIME25, GPQA) primarily test long-generation reasoning. However, many real-world LLM applications involve a much wider variety of scenarios, including short-output generation and general instruction-following. The test set should ideally cover a more diverse range of practical use cases to improve the applicability of the conclusions.

**Questions:**

1. The field is showing significant interest in Mixture-of-Experts (MoE) architectures. How do the authors anticipate that the difference between MoE models and Dense models would influence the conclusions presented in this paper?

---

> ### Author Response · Authors · 2025-11-21
> **Authors' Response to Reviewer EX7S**
>
> Thank you for your thoughtful review. We are glad that you appreciated our **exhaustive experiments** and **multi-dimensional analysis** of the complex trade-offs involved in model deployment, a strength also highlighted by the other reviewers (tfiR, Daoc). We also thank you for acknowledging that this approach has **direct and significant value** for real-world deployment scenarios.
>
> **W1. AIME25 may not be a suitable benchmark due to its extreme difficulty**
>
> We chose AIME25 deliberately **because of its difficulty** so that the benefits of test-time scaling are most clearly manifested. This ensures that small changes in deployment strategy lead to measurable differences in final accuracy, making it a suitable benchmark for evaluating memory–accuracy trade-offs in reasoning models. Furthermore, our work focuses on large reasoning models, which are often used for math and coding tasks. This is precisely why AIME25 is widely used in recent reasoning studies [1–5] to evaluate and compare model performance, underscoring its relevance as a representative benchmark.
>
> Regarding the reviewer's concern about high variance in results, we would like to emphasize that each problem is evaluated with 32 sampled solutions, and we report average accuracy. While variance remains even after averaging, **we avoid drawing conclusions from marginal differences and instead focus on stable accuracy gaps**. For example, when stating that the 1.7B model in 8-bit with a 6k-token budget outperforms the 0.6B model in 8-bit with an 18k-token budget (L239), the accuracy difference is **22.60% vs. 15.00%**, a clear and meaningful gap.
>
> **W2. Findings are specific to Qwen3 family**
>
> The reviewer raises a valid concern, which we also acknowledged as a limitation in Section 7. Conducting comparable analyses across multiple reasoning model families is challenging, as few currently provide a fine-grained scale ladder like the Qwen3 family.
>
> Nonetheless, to strengthen generalizability, we have extended our experiments to two additional model families, **DeepSeek-R1-Distill** and **OpenReasoning-Nemotron**, covering models from 1.5B to 14B parameters.
>
> ***OpenReasoning-Nemotron***
> |**Low-memory budget** (~6.8 GB)||**Acc. (%)**|**High-memory budget** (~31.0 GB)||**Acc. (%)**|
> |-|-|-:|-|-|-:|
> |**Model weights**|**KV cache**||**Model weights**|**KV cache**||
> |1.5B · 16-bit (2.91 GB)|18k tokens, g=8 (3.88 GB)|44.38|7B · 16-bit (14.27 GB)|26k tokens, g=12 (16.76 GB)|**81.14**|
> |7B · 4-bit (5.30 GB)|26k tokens, g=1 (1.40 GB)|**61.77**|14B · 16-bit (27.90 GB)|14k tokens, g=1 (2.59 GB)|52.50|
>
> ***DeepSeek-R1-Distill***
> |**Low-memory budget** (~9.6 GB)||**Acc. (%)**|**High-memory budget** (~30.1 GB)||**Acc. (%)**|
> |-|-|-:|-|-|-:|
> |**Model weights**|**KV cache**||**Model weights**|**KV cache**||
> |1.5B · 8-bit (2.12 GB)|18k tokens, g=16 (7.76 GB)|27.60|7B · 16-bit (14.19 GB)|18k tokens, g=16 (15.52 GB)|**54.52**|
> |7B · 8-bit (8.25 GB)|6k tokens, g=4 (1.31 GB)|**37.09**|14B · 16-bit (27.51 GB)|14k tokens, g=1 (2.59 GB)|39.17|
>
> In these tables, we compare, within matched memory budgets, pairs of configurations that allocate **more memory to the KV cache** (upper row) versus **more memory to model weights** (lower row) in both **low- and high-memory regimes**. Across both families, our main conclusion holds:
> > For smaller models, it is more memory-efficient to allocate memory to model weights; for larger models, to allocate memory to the KV cache for longer generations (L258).
>
> We also show the full landscape in **Figures 13 and 14** of the revised manuscript. For R1-Distill, parallel scaling of the 4-bit 7B model performs slightly below the Pareto frontier of serial scaling. For OpenReasoning-Nemotron, the 4-bit 7B model achieves a marginally stronger memory–accuracy trade-off. For both model families, models effectively smaller than this threshold show suboptimal memory trade-offs when forced to spend memory on the KV cache, while models effectively larger than the threshold benefit from parallel scaling. While the exact memory threshold shifts slightly by model family, the **existence of such a threshold** and the resulting memory-optimal scaling strategies remain *consistent* across reasoning model families.
>
> [1] Sadhukhan, Ranajoy, et al. "Kinetics: Rethinking Test-Time Scaling Laws." ArXiv, 2025
>
> [2] Sun, Yiyou, et al. "Climbing the Ladder of Reasoning: What LLMs Can-and Still Can't-Solve after SFT?." ArXiv, 2025
>
> [3] Wang, Yiping, et al. "Reinforcement learning for reasoning in large language models with one training example." ArXiv, 2025
>
> [4] Yu, Qiying, et al. "Dapo: An open-source llm reinforcement learning system at scale." ArXiv, 2025
>
> [5] Khatri, Devvrit, et al. "The art of scaling reinforcement learning compute for llms." ArXiv, 2025

---

> > ### Author Response · Authors · 2025-11-21
> > **Authors' Response to Reviewer EX7S**
> >
> > **W3. Limited task diversity**
> >
> > We first clarify that our study *intentionally* focuses on long CoT reasoning tasks. Some of the suggested scenarios, such as short-output generation and general instruction-following tasks, consume negligible KV cache memory. In contrast, our analysis targets long-generation scenarios, where the KV cache becomes the dominant memory factor. Consequently, the memory–accuracy trade-offs and optimal scaling policies in these two settings differ. Prior work [6,7,8] has **already established memory-optimal prescriptions for short-generation tasks**. Concretely, they recommend aggressive weight quantization near and below 4-bit when the KV cache is negligible. Our results complement these by extending the analysis to the long-generation regime, where the memory allocation between model weights and the KV cache plays a central role.
> >
> > To broaden diversity within reasoning tasks, we additionally evaluate on **LiveCodeBench** (code reasoning) and **MATH500** (a larger mathematical reasoning benchmark). Please see Figures 15 and 16 of the revised manuscript.
> >
> > For LiveCodeBench, we find a **trend very similar to AIME25**: for models with effective size below 8-bit 4B, allocating more memory to model weights yields larger gains, whereas above this threshold, memory is better spent increasing the test-time budget until performance saturates. We also show this in the table below, where we compare, within matched memory budgets, pairs of configurations that allocate **more memory to the KV cache** (upper row) versus **more memory to model weights** (lower row) in both low- and high-memory regimes. Also, unlike knowledge-intensive GPQA, **4-bit precision remains less memory-efficient** on LiveCodeBench: a 16-bit 4B configuration is strictly more memory-efficient than a 4-bit 14B configuration.
> >
> > ***LiveCodeBench (Qwen3)***
> > |**Low-memory budget** (~6.2 GB)||**Acc. (%)**|**High-memory budget** (~10.5 GB)||**Acc. (%)**|
> > |-|-|-:|-|-|-:|
> > |**Model weights**|**KV cache**||**Model weights**|**KV cache**||
> > |1.7B · 16-bit (3.78 GB)|22k tokens (2.43 GB)|42.50|4B · 16-bit (7.49 GB)|22k tokens (3.13 GB)|**65.00**|
> > |4B · 8-bit (4.19 GB)|14k tokens (2.03 GB)|**61.69**|14B · 4-bit (9.30 GB)|6k tokens (1.03 GB)|26.75|
> >
> > For MATH500, the **threshold shifts left** (toward smaller effective sizes) because the benchmark is easier than AIME25. Accuracy saturates quickly once models exceed roughly a 16-bit 1.7B effective size. However, for models below this memory threshold, investing memory in model weights is still more efficient than growing the KV cache, as shown in the table below. We also observe non-monotonicity in accuracy under budget forcing: for an already easy set, pushing generation length beyond what is needed can lead to reduced accuracy.
> >
> > ***MATH500 (Qwen3)***
> > |**Low-memory budget** (~3.0 GB)||**Acc. (%)**|**High-memory budget** (~7.2 GB)||**Acc. (%)**|
> > |-|-|-:|-|-|-:|
> > |**Model weights**|**KV cache**||**Model weights**|**KV cache**||
> > |0.6B · 8-bit (0.71 GB)|22k tokens (2.36 GB)|70.0|4B · 8-bit (4.19 GB)|22k tokens (3.04 GB)|**90.2**|
> > |1.7B · 8-bit (1.93 GB)|10k tokens (1.08 GB)|**83.0**|8B · 4-bit (5.68 GB)|10k tokens (1.39 GB)|87.8|
> >
> > Together, these results reinforce the **existence of a scale-dependent threshold** and the **corresponding memory-optimal strategies** on either side of it. They also strengthen our claim about numerical precision: reasoning in math and code is sensitive to low-bit settings, and higher precision can be more memory-efficient than scaling to a larger model at 4-bit precision under a fixed budget.
> >
> > **Q1. On MoE architectures**
> >
> > We appreciate the suggestion to consider Mixture-of-Experts (MoE) models. The main reason we did not include MoE models in this study is their **poor memory--accuracy trade-off by design**. MoE models must hold parameters for all experts in memory while activating only a subset for each forward pass. This design primarily aims to reduce compute cost per forward pass, leading to **strong compute and latency–accuracy trade-offs, but not improved memory efficiency**.
> >
> > That said, if one must deploy MoE, it would be valuable to study the memory–accuracy trade-off **within** a family of MoE models and determine the memory-optimal scaling strategy among MoE variants. At present, there is no publicly available MoE family with a fine-grained, consistent scale ladder suitable for such controlled experiments.
> >
> > [6] Dettmers, Tim, and Luke Zettlemoyer. "The case for 4-bit precision: k-bit inference scaling laws." ICML, 2023
> >
> > [7] Frantar, Elias, et al. "Gptq: Accurate post-training quantization for generative pre-trained transformers." ICLR, 2023
> >
> > [8] Chee, Jerry, et al. "Quip: 2-bit quantization of large language models with guarantees." NeurIPS, 2023

---

> > > ### Comment · Reviewer_EX7S · 2025-11-24
> > > **Thank you for your work on rebuttle, I will keep my score unchanged**
> > >
> > > Thank you for your work on rebuttle.
> > >
> > > 1. I agree with the argument that AIME, due to its difficulty, can be used to evaluate the impact of inference settings on performance. However, I reserve my opinion that more test sets should be added. Since this paper primarily tests real-world utility, it's crucial to consider that real-world users won't only use the inference model for inference tasks; more users will expect the inference model to perform better in general domains, including tasks like short generation and long-context reading comprehension. Including more results would strengthen the paper's conclusions.
> > >
> > > 2. I agree that the MoE model inherently has memory inefficiencies. However, since the most powerful existing models all use the MoE architecture, including analysis of MoE models (even just on a separate track) would be more meaningful for guiding real-world use. However, I also accept that this paper focuses only on Dense models and leaves comparisons of the MoE series for future work. (Furthermore, the Qwen3 series includes MoE models that can be used for comparison in future work).
> > >
> > > In conclusion, I accept most of the authors' explanations, but I will keep my score unchanged.

---

> > > > ### Author Response · Authors · 2025-11-28
> > > >
> > > > We thank the reviewer for their thoughtful follow-up and for engaging with our rebuttal. We are glad that many of our explanations were acceptable and that our use of AIME as a challenging benchmark is now clearer.
> > > >
> > > > **On the scope of evaluation benchmarks**
> > > >
> > > > As noted in our rebuttal (W3), we have expanded our evaluation beyond AIME25 (competition-style math, multi-step reasoning) and GPQA-Diamond (challenging scientific QA across physics, chemistry, and biology) to include LiveCodeBench (code reasoning) and MATH500 (a larger math benchmark).
> > > >
> > > > Across all these benchmarks (see tables under W3 and Figures 15 and 16), the core pattern remains consistent: there is a scale-dependent threshold where prioritizing KV cache over model weights becomes effective. This addition has satisfactorily addressed similar concerns from reviewers tfiR and Daoc.
> > > >
> > > > Regarding short-generation tasks, our work intentionally focuses on the long CoT regime where KV memory is dominant. For short generations, prior work [6,7,8] establishes that **aggressive weight quantization ($\leq$4 bits)** is memory-optimal. To confirm this, we compared an 8-bit 8B model against a 4-bit 14B model (both requiring ~9 GB for weights) on standard short-context benchmarks:
> > > >
> > > > | Model (precision, params) | ARC-Challenge | HellaSwag | LAMBADA | OpenBookQA | PIQA | Winogrande |
> > > > | :--- | :--- | :--- | :--- | :--- | :--- | :--- |
> > > > | 8-bit, 8B | 56.48% | 74.86% | 65.19% | 41.40% | 77.69% | 68.03% |
> > > > | **4-bit, 14B** | **60.58%** | **78.39%** | **67.84%** | **45.80%** | **79.92%** | **72.22%** |
> > > >
> > > > The 4-bit 14B model consistently wins, confirming that for short-context settings, the memory-optimal strategy is to use larger parameters with lower weight precision.
> > > >
> > > > **On including MoE models**
> > > >
> > > > We agree that a dedicated analysis of MoE models would provide valuable practical guidance. We are also encouraged by the reviewer’s acknowledgement that our dense model findings remain practically meaningful.
> > > >
> > > > A key barrier to including a separate "MoE track" is the lack of a publicly available MoE family with the fine-grained scale ladder required for our methodology. Our approach relies on sweeping nearby model sizes with overlapping memory budgets to isolate the trade-off between model weights and KV cache. Current Qwen3 MoE checkpoints have vast scale gaps (e.g., 30B vs. 235B), making such controlled comparisons infeasible.
> > > >
> > > > [6] Dettmers, Tim, and Luke Zettlemoyer. "The case for 4-bit precision: k-bit inference scaling laws." ICML, 2023
> > > >
> > > > [7] Frantar, Elias, et al. "GPTQ: Accurate post-training quantization for generative pre-trained transformers." ICLR, 2023
> > > >
> > > > [8] Chee, Jerry, et al. "Quip: 2-bit quantization of large language models with guarantees." NeurIPS, 2023

---

### Official Review · Reviewer_tfiR · 2025-11-03

**Soundness:** 4
**Presentation:** 4
**Contribution:** 3
**Rating:** 6
**Confidence:** 5

**Summary:**

The paper investigate a set of critical research questions to improve the ability of reasoning models given limited memory constraints. The authors find that the standard 4-bit quantization used for non-reasoning models fails for reasoning-focused Large Language Models (LLMs), as their memory is often dominated by the Key-Value (KV) cache rather than the model size itself . Through systematic experiments, the authors identify a scale-dependent trade-off: smaller models (below 8 billion effective parameters) achieve better accuracy by allocating memory to higher-precision weights, while larger models benefit from allocating memory to support longer generations . This leads to the central guideline that for small reasoning models, one should prioritize model capacity, but for larger ones, the focus should be on maximizing test-time compute . This scale threshold also determines when parallel scaling becomes efficient and whether KV cache eviction is a better strategy than KV quantization. While there are enumerous new compression algorithms proposed nowadays, it is critical to review these methods from a unified view, where the memory constraint could be a good dimension.

**Strengths:**

- The paper studies an important and fundamental research question: how to find the best strategy to improve the reasoning ability of the models given limited memory.

- The authors conduct solid experiments to verify each dimension of the problem: the precision, generation lengths, parallel sampling, KV-cache eviction, etc.

- The paper is well structured and easy to follow.  The main conclusions are properly highlighted and summarized.  Figures are nice to read.

**Weaknesses:**

Although the paper conducts solid experiments, I believe it is still a snapshot of the current situation, and the conclusion may still vary with different LLM families, tasks, or settings.  The paper would be more complete if the authors could provide results on

- more LLM families, such as the DeepSeek-R1-distilled-Qwen families, Seed-OSS 36B, etc.;

- more compression algorithms (e.g., weight–activation quantization or MXFP4/NVFP4 quantization, etc.).

In addition, it would also be helpful if the authors could compare the scale changes in more domains (especially non-reasoning tasks) under a limited-memory constraint.

**Questions:**

- How can the findings of this paper be applied in practice?  Usually the LLM is served in the cloud, where each batch of user queries has context from various domains.  The optimal scaling strategy could vary from task to task.

- How do you implement the parallel sampling with different group sizes?  Is it by Majority@K?
- Some recent related works could be incorporated.
  1. Li Z, Su Y, Yang R, et al.  Quantization meets reasoning: Exploring LLM low-bit quantization degradation for mathematical reasoning[J].  arXiv preprint arXiv:2501.03035, 2025.
  2. Liu R, Sun Y, Zhang M, et al.  Quantization hurts reasoning?  An empirical study on quantized reasoning models[J].  arXiv preprint arXiv:2504.04823, 2025.
  3. Zhang N, Zhang Y, Mitra P, et al.  When reasoning meets compression: Benchmarking compressed large reasoning models on complex reasoning tasks[J].  arXiv preprint arXiv:2504.02010, 2025.

---

> ### Author Response · Authors · 2025-11-21
> **Authors' Response to Reviewer tfiR**
>
> We thank the reviewer for the thoughtful and constructive feedback. We are pleased that you appreciated the **importance of our research question**: how to find the best strategy to improve models’ reasoning ability given limited memory. It is also encouraging that all three other reviewers similarly highlighted the significance and practicality of our findings. We also thank you for recognizing our **solid experimental design**, which covers different dimensions of the problem, and the **organization of the paper**.
>
> **W1. Provide results on more LLM families**
>
> Conducting comparable analyses across many model families is challenging because few reasoning model families currently provide a fine-grained scale ladder like Qwen3. Nonetheless, we have conducted additional experiments on two other model families: **DeepSeek-R1-Distill** and **OpenReasoning-Nemotron** (1.5B–14B) on AIME25.
>
> ***OpenReasoning-Nemotron***
> |**Low-memory budget** (~6.8 GB)||**Acc. (%)**|**High-memory budget** (~31.0 GB)||**Acc. (%)**|
> |-|-|-:|-|-|-:|
> |**Model weights**|**KV cache**||**Model weights**|**KV cache**||
> |1.5B · 16-bit (2.91 GB)|18k tokens, g=8 (3.88 GB)|44.38|7B · 16-bit (14.27 GB)|26k tokens, g=12 (16.76 GB)|**81.14**|
> |7B · 4-bit (5.30 GB)|26k tokens, g=1 (1.40 GB)|**61.77**|14B · 16-bit (27.90 GB)|14k tokens, g=1 (2.59 GB)|52.50|
>
> ***DeepSeek-R1-Distill***
> |**Low-memory budget** (~9.6 GB)||**Acc. (%)**|**High-memory budget** (~30.1 GB)||**Acc. (%)**|
> |-|-|-:|-|-|-:|
> |**Model weights**|**KV cache**||**Model weights**|**KV cache**||
> |1.5B · 8-bit (2.12 GB)|18k tokens, g=16 (7.76 GB)|27.60|7B · 16-bit (14.19 GB)|18k tokens, g=16 (15.52 GB)|**54.52**|
> |7B · 8-bit (8.25 GB)|6k tokens, g=4 (1.31 GB)|**37.09**|14B · 16-bit (27.51 GB)|14k tokens, g=1 (2.59 GB)|39.17 |
>
> In these tables, we compare, within matched memory budgets, pairs of configurations that allocate **more memory to the KV cache** (upper row) versus **more memory to model weights** (lower row) in both **low- and high-memory regimes**. Across both families, our main conclusion holds:
>
> > For smaller models, it is more memory-efficient to allocate memory to model weights; for larger models, to allocate memory to the KV cache for longer generations (L258).
>
> We also show the full landscape in **Figures 13 and 14** of the revised manuscript. For R1-Distill, parallel scaling of the 4-bit 7B model performs slightly below the Pareto frontier of serial scaling. For OpenReasoning-Nemotron, the 4-bit 7B model achieves a marginally stronger memory–accuracy trade-off. For both model families, models effectively smaller than this threshold show suboptimal memory trade-offs when forced to spend memory on the KV cache, while models effectively larger than the threshold benefit from parallel scaling. While the exact memory threshold shifts slightly by model family, the **existence** of such a threshold and the resulting memory-optimal scaling strategies remain *consistent* across reasoning model families.
>
> **W2. Provide results on more compression algorithms**
>
> We appreciate this suggestion to strengthen the completeness of our analysis. To test whether our conclusions depend on a particular quantization scheme, we replicated the analysis using **AWQ** and **FP8** quantization on 1.7B, 4B, and 8B models.
>
> When plotted alongside GPTQ, as shown in **Figure 17** of the revised draft, the memory–accuracy curves from AWQ and FP8 nearly overlap, confirming that the observed trend is not tied to a specific quantization algorithm. Furthermore, we show that our key finding holds consistently across these quantization methods.
>
> |**Low-memory budget** (~4.1 GB)||**Acc. (%)**|**High-memory budget** (~8.3 GB)||**Acc. (%)**|
> |-|-|-:|-|-|-:|
> |**Model weights**|**KV cache**||**Model weights**|**KV cache**||
> |1.7B · FP8 (1.92 GB)|22k tokens, g=1 (2.37 GB)|36.67|4B · FP8 (4.23 GB)|30k tokens, g=1 (4.14 GB)|**56.04**|
> |4B · AWQ-4-bit (2.51 GB)|10k tokens, g=1 (1.39 GB)|**45.62**|8B · AWQ-4-bit (5.71 GB)|18k tokens, g=1 (2.49 GB)|46.46|
>
> At a comparable memory budget of around **4.1 GB**, a smaller model with a longer generation, the **FP8 1.7B model with 22k tokens (36.7%)**, performs worse than a larger model with a shorter generation, the **AWQ-4-bit 4B model with 10k tokens (45.6%)**. Conversely, at a higher memory budget of around **8 GB**, the relationship reverses: the **FP8 4B model with 30k tokens (56.0%)** outperforms the larger model but shorter-generation **AWQ-4-bit 8B model with 18k tokens (46.5%)**.
>
> Overall, these results confirm our key conclusion: **for smaller models, allocate memory to model weights; for larger models, allocate memory to the KV cache to support longer generations**. This finding is robust across different quantization algorithms and is not tied to GPTQ.

---

> > ### Author Response · Authors · 2025-11-21
> > **Authors' Response to Reviewer tfiR**
> >
> > **W3. Provide results on more task domains, especially non-reasoning tasks**
> >
> > We would like to emphasize that our study *intentionally* focuses on long CoT reasoning tasks, where KV cache memory becomes a significant deployment consideration. This distinguishes our setting from non-reasoning tasks, in which the KV cache size is often negligible. Prior work [1,2,3] has **already established memory-optimal prescriptions for short-generation tasks**, specifically, recommending aggressive weight quantization near and below 4-bit when the KV cache is negligible. Our results complement these by extending the analysis to the long-generation regime, where the allocation between model weights and the KV cache plays a central role.
> >
> > As suggested by the reviewer, to broaden the evaluation within long reasoning, we include two additional benchmarks: **LiveCodeBench** for reasoning in code generation and **MATH500** for a larger and more diverse set of math questions. Please see Figures 15 and 16 of the revised manuscript.
> >
> > For LiveCodeBench, we find a **trend very similar to AIME25**: for models with effective size below 8-bit 4B, allocating more memory to model weights yields larger gains, whereas above this threshold, memory is better spent increasing the test-time budget until performance saturates. We also show this in the table below, where we compare, within matched memory budgets, configurations that allocate **more memory to the KV cache** (upper row) versus **more memory to model weights** (lower row) in both low- and high-memory regimes. Also, unlike knowledge-intensive GPQA, **4-bit precision remains less memory-efficient** on LiveCodeBench: a 16-bit 4B configuration is strictly more memory-efficient than a 4-bit 14B configuration.
> >
> > ***LiveCodeBench (Qwen3)***
> > |**Low-memory budget** (~6.2 GB)||**Acc. (%)**|**High-memory budget** (~10.5 GB)||**Acc. (%)**|
> > |-|-|-:|-|-|-:|
> > |**Model weights**|**KV cache**||**Model weights**|**KV cache**||
> > |1.7B · 16-bit (3.78 GB)|22k tokens (2.43 GB)|42.50|4B · 16-bit (7.49 GB)|22k tokens (3.13 GB)|**65.00**|
> > |4B · 8-bit (4.19 GB)|14k tokens (2.03 GB)|**61.69**|14B · 4-bit (9.30 GB)|6k tokens (1.03 GB)|26.75|
> >
> > For MATH500, the **threshold shifts left** (toward smaller effective sizes) because the benchmark is easier than AIME25. Accuracy saturates quickly once models exceed roughly a 16-bit 1.7B effective size. However, for models below this memory threshold, investing memory in model weights is still more efficient than growing the KV cache, as shown in the table below. We also observe non-monotonicity in accuracy under budget forcing: for an already easy set, pushing generation length beyond what is needed can reduce accuracy.
> >
> > ***MATH500 (Qwen3)***
> > |**Low-memory budget** (~3.0 GB)||**Acc. (%)**|**High-memory budget** (~7.2 GB)||**Acc. (%)**|
> > |-|-|-:|-|-|-:|
> > |**Model weights**|**KV cache**||**Model weights**|**KV cache**||
> > |0.6B · 8-bit (0.71 GB)|22k tokens (2.36 GB)|70.0|4B · 8-bit (4.19 GB)|22k tokens (3.04 GB)|**90.2**|
> > |1.7B · 8-bit (1.93 GB)|10k tokens (1.08 GB)|**83.0**|8B · 4-bit (5.68 GB)|10k tokens (1.39 GB)|87.8|
> >
> > Together, these results reinforce the **existence of a scale-dependent threshold** and the **corresponding memory-optimal strategies** on either side of it. They also strengthen our claim about numerical precision: reasoning in math and code is sensitive to low-bit settings, and higher precision can be more memory-efficient than scaling to a larger model at 4-bit precision under a fixed budget.
> >
> > [1] Dettmers, Tim, and Luke Zettlemoyer. "The case for 4-bit precision: k-bit inference scaling laws." ICML, 2023
> >
> > [2] Frantar, Elias, et al. "Gptq: Accurate post-training quantization for generative pre-trained transformers." ICLR, 2023
> >
> > [3] Chee, Jerry, et al. "Quip: 2-bit quantization of large language models with guarantees." NeurIPS, 2023

---

> > > ### Author Response · Authors · 2025-11-21
> > > **Authors' Response to Reviewer tfiR**
> > >
> > > **Q1. How can the findings be applied in practice?**
> > >
> > > Our findings are highly relevant for *on-device* deployments. On-device language model usage now represents a significant and growing share of real-world inference scenarios, as seen in [4–7] and references therein. These applications typically operate with a **single batch size**, face **strict memory limits**, and target a **specific task domain**. Such settings directly align with our main analysis.
> > >
> > > When LLMs are served with multi-batching, the cost of model weights is amortized, and the significance of the KV cache becomes even more amplified. Appendix C.2 provides insight into such settings, and many of our main findings apply to batched inference as well. For instance, our takeaways on KV cache compression (Findings 4 and 5) remain valid regardless of the deployment setting.
> > >
> > > Furthermore, in practical large-scale deployments, we expect that grouping requests from similar task domains could enable adaptive scaling strategies tuned to each task type. While developing such scheduling policies is beyond the current scope, we view it as an important direction for future work.
> > >
> > > **Q2. How is parallel sampling implemented? Is it by Majority@K?**
> > >
> > > Yes. We implement parallel sampling by generating multiple independent reasoning trajectories at temperature 0.6 and selecting the final answer through majority voting, where the most frequent answer among the sampled outputs is chosen as the final prediction. This procedure is described in L159-161 and L317-319.
> > >
> > > **Q3. Incorporate recent related works**
> > >
> > > Thank you for pointing us to recent related works. We have incorporated a relevant discussion of these studies in the revised draft. (L740-748) To summarize the discussion that has been added: Recent studies examine how compression and low-bit quantization affect reasoning. [8] finds that aggressive weight quantization notably harms mathematical reasoning at low precision, consistent with our observation that 4-bit precision is memory-inefficient for mathematical reasoning; [9] shows the effect varies by bit-width and model family; [10] benchmarks compressed reasoning models on complex tasks to chart accuracy under compression. We reframe the problem as selecting a memory-optimal strategy for reasoning, identify a scale-dependent threshold for allocating memory to model weights versus longer generations, and incorporate KV cache compression into our analysis.
> > >
> > > [4] Saad-Falcon, Jon, et al. "Intelligence per Watt: Measuring Intelligence Efficiency of Local AI." ArXiv, 2025
> > >
> > > [5] Qu, Guanqiao, et al. "Mobile edge intelligence for large language models: A contemporary survey." IEEE Communications Surveys & Tutorials, 2025
> > >
> > > [6] Zheng, Yue, et al. "A review on edge large language models: Design, execution, and applications." ACM Computing Surveys, 2025
> > >
> > > [7] Liu, Zechun, et al. "Mobilellm: Optimizing sub-billion parameter language models for on-device use cases." ICML, 2024
> > >
> > > [8] Li, Zhen, et al. "Quantization meets reasoning: Exploring llm low-bit quantization degradation for mathematical reasoning." ArXiv, 2025
> > >
> > > [9] Liu, Ruikang, et al. "Quantization hurts reasoning? an empirical study on quantized reasoning models." ArXiv, 2025
> > >
> > > [10] Zhang, Nan, et al. "When reasoning meets compression: Benchmarking compressed large reasoning models on complex reasoning tasks." ArXiv, 2025

---

> ### Comment · Reviewer_tfiR · 2025-11-27
>
> I appreciate the authors detailed response to my questions. I think most of my concerns are addressed and I would keep my original evaluation for accepting this paper.

---

> > ### Author Response · Authors · 2025-11-28
> >
> > We thank you for your careful reading of our paper and your positive evaluation. We are glad that our responses have addressed most of your concerns, and we very much appreciate your questions and suggestions, which will help us further improve the paper. If you have any remaining questions or additional suggestions, we would be very happy to address them.

---

### Comment · Area_Chair_wi7d · 2025-11-29

I appreciate the authors for all the rebuttal and additional experiments. However, when I checked the revised manuscript, most add-ons were put in the appendix. Since ICLR allows [an additional page](https://iclr.cc/Conferences/2026/AuthorGuide) in the rebuttal phase (max to 10 pages), could you let me know how you will accommodate the reviewers' comments into your revised manuscript?

---

> ### Author Response · Authors · 2025-11-30
>
> Thank you for pointing out the 10-page limit and for carefully reviewing both our paper and rebuttal. In our initial rebuttal draft, we placed most of the new material in the appendix so that reviewers could easily cross-reference detailed plots and tables.
>
> In the revised manuscript, we have now moved the key improvements requested by the reviewers into the main body while staying within the 10-page limit. All additions made during the rebuttal phase are marked in blue in the PDF.
>
> Below is a summary of how we have incorporated the reviewers’ comments into the revised manuscript:
>
> * **Introduction, background, experimental setup, and limitations updated:** We revised these sections to reflect the newly added experiments, describing the additional model families, benchmarks, and quantization schemes, and softening the stated limitations accordingly.
> * **Generalization across quantization methods:** At the beginning of Section 4 (L264-267), we added a brief discussion showing that our key memory–accuracy conclusions are robust across GPTQ, AWQ, and FP8, and we link to the more detailed analysis in Appendix C.2.
> * **LiveCodeBench and MATH500 results:** We introduced LiveCodeBench as a new main-text figure (Figure 3) with accompanying discussion, and we added a pointer to Appendix C.4, where we provide a more detailed analysis of LiveCodeBench and MATH500 results (L309-311).
> * **DeepSeek-R1-Distill and OpenReasoning-Nemotron results:** We added DeepSeek-R1-Distill results as a main-text figure (Figure 6) with discussion showing that the scale-dependent behavior persists for different model families, and we link to Appendix C.6 for a more comprehensive analysis that also includes OpenReasoning-Nemotron results (L365-369).
> * **New subsection on external verifier experiments:** In the main body, we added a subsection on "Parallel Scaling with an External Verifier" (Section 4.1), presenting the Best-of-N + ActPRM-X experimental results and discussing why external verifiers are memory-inefficient compared to majority-vote parallel scaling (L378-390).
> * **Related work updated:** The related work section has been expanded to cover additional recent work on quantization and reasoning models suggested by reviewer tfiR (L794-802).
>
> Thank you again for the prompt and careful review of our paper and rebuttal. We are happy to address any further questions you may have.

---

### Meta-Review · Area_Chair_wi7d · 2026-01-07

**Summary:**

Across reviews, there is broad agreement on the importance of the research question identified in this paper, which provides a useful empirical characterization of memory–accuracy trade-offs for reasoning-time scaling under a fixed memory budget. Reviewers consistently praise the clean and transparent memory accounting, the systematic multi-dimensional grid and the clarity of presentation. The common concerns center on generalizability, baseline completeness, and theoretical justification of the threshold. Overall, the paper presents substantial empirical evidence with transparent accounting and yields actionable deployment guidance for reasoning workloads where long generations make KV cache a first-order memory bottleneck. The “8-bit 4B” value should be interpreted as an empirically observed inflection region under the evaluated settings, but the core five findings raised in this paper are well supported.

**Reviewer Concerns:**

There are a couple of common concerns regarding the generalizability beyond Qwen3 and broader benchmark coverage beyond long-form reasoning.
- Reviewer EX7S and Daoc are concerned that all evaluation results are confined to Qwen3 and present a narrow evaluation scope. This was substantially addressed with the results of DeepSeek-R1-Distill and OpenReasoning-Nemotron.
- EX7S and SZ2z both pushed for broader task diversity / more practical coverage. This was partially addressed with additional LiveCodeBench and MATH500, but the evaluation is still primarily reasoning-centric, and the limitations section explicitly notes the scope is intentionally focused and broader comparisons are future work.
- Reviewer tfiR asked for more compression algorithms beyond the explored set. This was partially addressed by the authors with more results of AWQ and FP8.
- Reviewer Daoc pointed out the weak theoretical mechanistic explanation of the thresholding. The authors pin-pointed a discussion and defended that the result was an empirical scaling-style finding.

**Reviewer Scores:**

Reviewer Daoc explicitly mentioned their tendency to raise the final score (probably 6 to 8) on Nov 27. Two other reviewers decided to keep their score, and one remains non-responsive.

---

### Decision · Program_Chairs · 2026-01-26

Accept (Poster)